# No Train, all Gain: Self-Supervised Gradients Improve Deep Frozen Representations

Walter Simoncini[1,*]     Spyros Gidaris[2]     Andrei Bursuc[2]     Yuki M. Asano[1]

[1]QUVA Lab, University of Amsterdam; [2]valeo.ai, Paris, France

## Abstract

This paper introduces FUNGI, **F**eatures from **UN**supervised **G**rad**I**ents, a method to enhance the features of transformer encoders by leveraging self-supervised gradients. Our method is simple: given any pretrained model, we first compute gradients from various self-supervised objectives for each input. These gradients are projected to a lower dimension and then concatenated with the model's output embedding. The resulting features are evaluated on k-nearest neighbor classification over 11 datasets from vision, 5 from natural language processing, and 2 from audio. Across backbones spanning various sizes and pretraining strategies, FUNGI features provide consistent performance improvements over the embeddings. We also show that using FUNGI features can benefit linear classification, clustering and image retrieval, and that they significantly improve the retrieval-based in-context scene understanding abilities of pretrained models, for example improving upon DINO by +17% for semantic segmentation – without any training. Code is available at https://github.com/WalterSimoncini/fungivision.

## 1   Introduction

The k-nearest neighbor algorithm (kNN) (Fix, 1985) is a fundamental non-parametric machine learning tool, and can be scaled to datasets with billion of examples thanks to advances in quantization (Jegou et al., 2010; Guo et al., 2020) and efficient GPU implementations (Johnson et al., 2019). This simple and versatile algorithm has shown potential in multiple applications well before deep neural networks became relevant (Efros & Leung, 1999; Hays & Efros, 2008; Torralba et al., 2008). Its recent applications include fast and robust image classification with Vision Transformers (Caron et al., 2021; Chen & He, 2021), unlabeled data selection (Yalniz et al., 2019), relevant text-retrieval (Lewis et al., 2020), and visual in-context learning (Balazevic et al., 2024), where a context of data samples with their annotations (e.g., a semantic segmentation map) are used to make dense predictions.

Devising powerful and expressive features for recognition and image understanding has a long history in computer vision. Feature engineering strategies range from simple local features (Lowe, 2004; Dalal & Triggs, 2005; Van De Sande et al., 2009) extracting gradient, boundary or color information, to various mid-level (Boureau et al., 2010) or global pooling (Oliva & Torralba, 2001; Sivic & Zisserman, 2003; Jégou et al., 2010) techniques. It is also possible to couple off-the-shelf pretrained backbones as feature extractors with such pooling strategies (Gong et al., 2014; Kulkarni et al., 2015; Gidaris et al., 2020) to improve performances. While these approaches demonstrate the utility of using a neural network's learned embedding space, they still require specific expertise and tuning for each backbone and task with only limited guidance from the data itself.

---

*Work partially done as an intern at valeo.ai. Datasets were solely downloaded and evaluated by the University of Amsterdam. walter@ashita.nl

38th Conference on Neural Information Processing Systems (NeurIPS 2024).

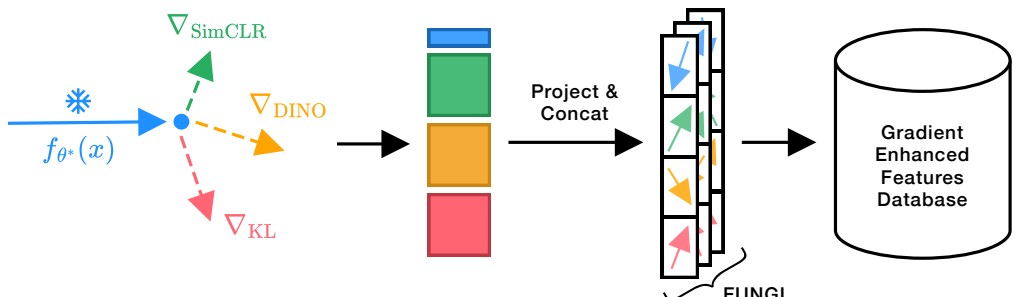

Figure 1: **Gradient-augmented features**: given a pretrained backbone $f_{\theta^*}$ and its embeddings, we apply a family of SSL losses, extract their gradients, and project and concatenate them. These new features are used to build a $k$-nearest neighbor index, which can be used for classification or retrieval.

We depart from this line of work and aim to attain strong representations without training and feature engineering, yet still exploiting information cues from data. In particular, we suggest to enhance the neural network's embeddings by incorporating FUNGI (Features from Unsupervised GradIents). FUNGI are obtained from self-supervised loss functions, as these do not require any human annotations and allow for a simple enhancement to embedding-only kNN. The losses are computed on top of pretrained backbones (with randomly initialized linear layers if needed), which permits our method to be "plug-and-play" and benefit from the diverse set of pretraining objectives put forth by the community.

We explore gradients from various learning objectives, such as contrastive learning (Chen et al., 2020a) and self-distillation (Caron et al., 2021) thereby integrating complementary information that mitigates the weaknesses of individual losses. The gradients are obtained from late hidden layers, making them computationally cheap. Finally, these are projected to smaller dimensions, and concatenated with the neural network embeddings, to yield new inputs to the classic kNN algorithm.

Using kNN with FUNGI can be regarded as a non-parametric transfer learning approach: the gradient information encodes first-order learning signals that are specific to the downstream data, yet no parameters need to be updated. Despite this simplicity, we achieve consistent performance improvements across multiple models and benchmarks. Overall, our main contributions are summarized as follows:

- We introduce FUNGI, a novel method that combines neural network features and gradients to enhance representations.
- We demonstrate that the gradients from self-supervised losses have predictive abilities and offer complementary information to model embeddings.
- We validate the generality and utility of our method by achieving consistent gains across 11 image, 5 text, and 2 audio classification benchmarks, plus 2 in-context image segmentation and 2 image retrieval tasks, utilizing a total of 20 backbones.

## 2 Related Work

**Fast Adaptation** There is a broad range of approaches to quickly adapt models to newly specified tasks and data. Inspired by early *learning-to-learn* work (Hochreiter et al., 2001), *meta-learning* methods (Finn et al., 2017; Nichol et al., 2018) learn to initialize the parameters of a learner such that it becomes faster to fine-tune with a small number of gradient steps and data. Alternative approaches leverage external memory modules to store relevant training samples to learn to match query examples (Santoro et al., 2016; Vinyals et al., 2016), learn to produce and compare class-based prototypes (Snell et al., 2017) or learn to generate the weights of a classifier (Gidaris & Komodakis, 2018) or even of an entire neural network (Bertinetto et al., 2016) from only a few labeled examples. The advent of Vision Transformers (Dosovitskiy et al., 2021) enable new parameter- and data-efficient strategies to adapt pretrained models through visual prompts (Jia et al., 2022) and in-context learning (Zhang et al., 2023). In contrast to this line of works, our method does not require specialized training and can be applied to any frozen pretrained backbone. FUNGI can also be related to test-time training (Sun et al., 2020; Hardt & Sun, 2024), where the parameters of a predictive model are

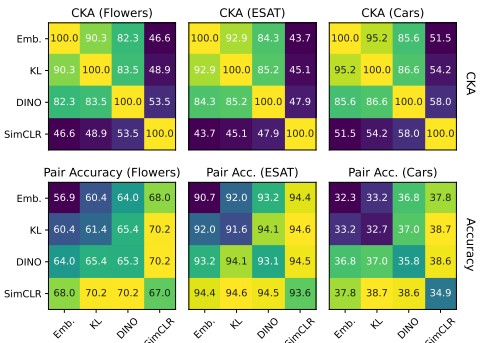

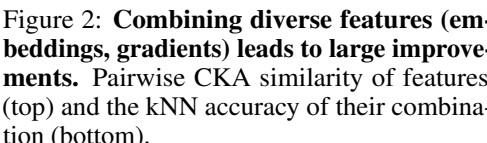

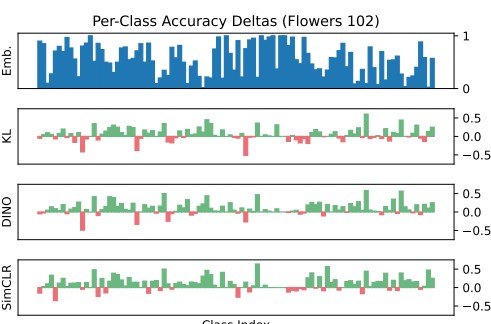

Figure 2: **Combining diverse features (embeddings, gradients) leads to large improvements.** Pairwise CKA similarity of features (top) and the kNN accuracy of their combination (bottom).

Figure 3: **Gradients encode different information.** Delta in per-class kNN accuracy of gradients from different objectives compared to the embeddings, indicated as "Emb." in the plot.

updated over test samples with a self-supervised objective to reduce the gap between the training and test distributions. While we also use self-supervised objectives and gradients, our approach does not update model parameters and is not limited to predictive models, as it can be applied to any task that can be solved with retrieval.

**Self-Supervised Learning Objectives** In recent years, self-supervised learning (SSL) has made tremendous progress in computer vision. SSL aims to learn good representations from unlabeled data by leveraging supervision from different signals in the data itself via pretext objectives, thus foregoing human supervision. Models pretrained with self-supervision are subsequently finetuned to downstream tasks of interest with few labeled samples. The crux of SSL is in the pretext learning objective. A wide and diverse collection of pretext objectives have been proposed in the community relying on contrastive learning (Chen et al., 2020a; He et al., 2020; Chen et al., 2020b), clustering (Caron et al., 2018; Asano et al., 2020; Caron et al., 2020), self-distillation (Caron et al., 2021; Grill et al., 2020; Chen & He, 2021; Gidaris et al., 2021), feature (Zhou et al., 2022; Assran et al., 2023) or input reconstruction (He et al., 2022).

We hypothesize that the gradients induced by these objectives encapsulate different information from the input data, and that this information can be combined to produce more information-rich representations. Here, we do not use self-supervision in the usual way, *i.e.*, to pretrain an encoder, but rather focus on pretext objectives and data augmentation strategies to compute representations from a frozen pretrained model.

**Feature Engineering** A long-standing research area for pattern recognition and image understanding before the advent of deep neural networks that brought the paradigm of end-to-end representation learning. In contrast, classic feature extraction methods are devised without labeled data and often from only a few data samples. They range from local features, such as SIFT (Lowe, 2004), HOG (Dalal & Triggs, 2005), to global pooling, such as GIST (Oliva & Torralba, 2001), Bag-of-Visual-Words (Sivic & Zisserman, 2003), Fisher vectors (Perronnin et al., 2010), VLAD (Jégou et al., 2010), selective match kernels (Tolias et al., 2013), etc. These pooling strategies can be easily plugged to intermediate or output neural network activations (Gong et al., 2014; Kulkarni et al., 2015; Gidaris et al., 2020), harnessing data-driven learned representations. Other modern examples of feature engineering include Head2Toe (Evci et al., 2022), which augments the model embeddings using intermediate activations, kNN-prompting (Xu et al., 2023), which uses the next token probabilities of a language model to perform few shot nearest neighbor classification and LOST (Siméoni et al., 2021) which uses patch features from self-supervised vision transformers for object localization. Closer to our line of work, Wei et al. (2022) shows that kernel regression using the empirical neural tangent kernel (eNTK), which corresponds to the model Jacobian, can achieve a performance similar to fine-tuning in vision, and Mu et al. (2020) shows that features obtained from the Jacobian of the top layers of a convolutional neural network can be used to enhance a linear classifier. In contrast, our method does not require any annotations and is computationally cheaper, as we only compute the gradient for a single layer rather than the full Jacobian.

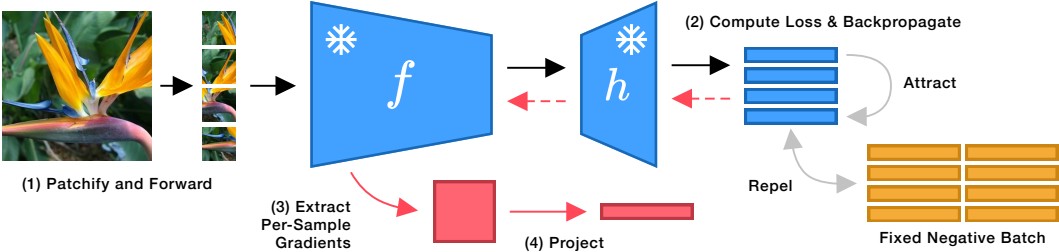

Figure 4: **Gradients extraction using a SimCLR loss.** Given a pretrained backbone $f$ and a randomly initialized projection head $h$, we first patchify an image, obtain the latent representations of patches (1), calculate the SimCLR loss by maximizing the pairwise cosine similarity of patches, and minimizing their similarity to a fixed negatives batch and backpropagate (2), extract the per-sample gradients (3) and finally project the gradients to the same dimensionality as the embeddings (4).

## 3 Gradients as Features

Gradients encode information on how the weights of a neural network should change to solve a given task. Thus, they contain information about the current state of the network and its relation to the data point(s) used to compute it. Therefore, we hypothesize that features from self-supervised gradients should perform better than the embeddings, as they are adapted to the dataset at hand. Empirically, the second row in Figure 2 shows this to be accurate, e.g., gradients from a SimCLR loss improve the accuracy of k-nearest neighbor classification by 10.1% on Flowers102.

If we plot the per-class accuracy distribution as in Figure 3, we notice that gradient features encode different information depending on the loss and that they can perform significantly worse on some classes, possibly because they are estimated using a single data point, and are thus dependent on the local loss curvature. These findings suggest that the information in embeddings and gradient features could be complementary. We show that this holds in the second row of Figure 2, as feature pairs perform better. Moreover, the first row of the figure suggests that more diverse feature pairs, as measured via their centered kernel alignment (CKA) (Kornblith et al., 2019) score, lead to better overall performance.

## 4 Method

Our method, FUNGI, enhances k-nearest neighbor search by incorporating features from unsupervised gradients. We extract gradients from self-supervised loss functions, project them to smaller dimensions, and concatenate them with neural network embeddings. The extraction of self-supervised gradients is illustrated in Figure 4, while Figure 1 shows how FUNGI features are constructed.

**Definitions**    Throughout this section, we define $L_2$ normalization as $z' = z/||z||_2$, a vision backbone as $f$, a linear projection head as $h$ and vectorization as $\mathtt{vec}(\cdot)$.

### 4.1 FUNGI: Features from Unsupervised Gradients

**Gradients Extraction.**    Given an arbitrary vision backbone $f$, in our case a vision transformer (ViT) (Dosovitskiy et al., 2021), we attach a randomly initialized linear projection head $h$ and obtain a latent representation $z = h(f'(x))$ of the input images, which we use to compute the loss for one of our self-supervised objectives. We then run backpropagation and extract the gradients with respect to the weights and biases of an arbitrary hidden linear layer within $f$. Unless specified otherwise, we use the attention output projection of the last transformer block as our gradient's source.

**From Gradients to Retrieval-Friendly Features.**    Gradients are high dimensional and thus impractical for nearest-neighbor retrieval due to speed and storage considerations and the curse of dimensionality. To tackle these issues, we downsample the gradients to the dimensionality of original model embeddings $d$ using the binary random projections method introduced by Achlioptas (2003). For this, we first vectorize the gradients by flattening them to a $m$-dimensional vector and then multiply them by a matrix $R \in \{-1, 1\}^{d,m}$ whose entries are the realizations of a Bernoulli random

variable with $p = 0.5$. The gradient $g_\beta$ with respect to a loss $\mathcal{L}_\beta$ is then defined as

$$g_\beta(x) = R\, \texttt{vec}\left(\nabla \mathcal{L}_\beta(x)\right). \tag{1}$$

**Combining with Embeddings.** To produce our FUNGI features $\phi$, we concatenate one or more gradients to the model embeddings. As gradient magnitudes can vary widely across losses, and we want gradients to be equally considered as the embeddings, we $L_2$-normalize each gradient, as well the embeddings and compute

$$\phi(x) = \texttt{cat}\left[g'_{\beta_1}(x), g'_{\beta_2}(x), ..., f'(x)\right], \tag{2}$$

where $\texttt{cat}$ denotes concatenation. Finally, we reduce the dimensionality of these combined features via PCA to a $d$-dimensional vector. This allows the combination of multiple losses at iso-storage cost. Our final FUNGI features for a sample $x$ are thus obtained as:

$$\phi_{\text{PCA}}(x) = \texttt{PCA}_d\left(\phi(x)\right). \tag{3}$$

## 4.2 Self-Supervised Objectives

We consider losses representing three families of self-supervised objectives: DINO (Caron et al., 2021), SimCLR (Chen et al., 2020a) and a KL-divergence based loss inspired by the *out-of-distribution* detection literature (Huang et al., 2021). In this section we briefly describe the objectives and our adjustments to them, and in Appendix B.6, we also briefly discuss clustering and masked image modeling-based losses.

**DINO.** DINO is a distillation and implicit clustering-based learning method. We use the standard DINO loss, which, given an image, enforces global and local crop correspondence between teacher and student models using a cross-entropy loss. In our case, both models share the same parameters, but have independent heads $h_s$ and $h_t$ for student and teacher respectively, thus we have $z_i = h_i\left(f'(x)\right), i \in \{s, t\}$. The DINO objective can be expressed as:

$$\mathcal{L}_{\text{DINO}} = \texttt{Cross-Entropy}\left(z_s, z_t\right). \tag{4}$$

**SimCLR.** SimCLR is a noise-contrastive method. Given a batch of images, SimCLR generates two views for each image and aims to minimize the distance between views belonging to the same image and maximize their distance to all other views. Instead, we generate a set of 49 overlapping patches for each image, which we call the positive set. This set is then contrasted against a fixed comparison batch of $49 \times 256$ negative examples. Our objective is the expectation of the pair-wise InfoNCE (Oord et al., 2018) loss for each pair of positive views. If we define the positive set of latent view representations as $Z$, where $z_i \in Z = h'(f(x_i))$ for a view $x_i$, the comparison batch size as $N$ and the temperature as $\tau$, the $\mathcal{L}_{\text{SimCLR}}$ objective is then defined as:

$$\mathcal{L}_{\text{SimCLR}} = \mathbb{E}_{(z_i, z_j) \sim Z, z_i \neq z_j}[\ell_{z_i, z_j}] \qquad \ell_{z_i, z_j} = -\log \frac{\exp(\text{sim}(z_i, z_j)/\tau)}{\sum_{k=1}^{49(N+1)} \mathbb{1}_{[k \neq i]} \exp(\text{sim}(z_i, z_k)/\tau)}. \tag{5}$$

**KL Divergence.** The KL objective is calculated as the KL divergence between the softmaxed logits of the latents and a uniform distribution $\mathcal{U}$:

$$\mathcal{L}_{\text{KL}} = \text{KL}(\text{softmax}(z)||\mathcal{U}). \tag{6}$$

We hypothesize two reasons as for why this loss produces predictive gradients: first, it receives a non-augmented image, with a higher signal-to-noise ratio compared to other objectives, and second, if we assume that similar images (*e.g.*, the ones that belong to the same class) produce similar activations, then maximizing their entropy by forcing the output distribution to match an uniform should produce similar intra-class gradients and help separability. This hypothesis is supported by the fact that while the KL gradients are discriminative, they have chance performance in other tasks, such as in-context scene understanding.

## 4.3 In-Context Scene Understanding

Balazevic et al. (2024) introduced a method for retrieval-based in-context scene understanding, where, for semantic segmentation, they first build a memory bank containing training image patches and their labels, and at test time, for each image patch, retrieve its nearest neighbors and use them to predict its

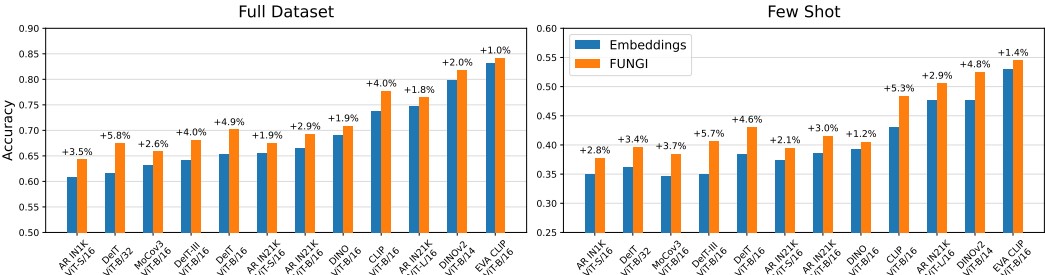

Figure 6: **FUNGI works across backbones.** Accuracy in $k$-nearest neighbor classification using embeddings and FUNGI features from various ViT backbones, both for full dataset and few shot setups, averaged over 11 datasets. For the FUNGI features we chose the best performing combination across datasets. "AR" indicates backbones trained with the AugReg strategy (Steiner et al., 2022).

Table 1: **FUNGI features are better on several datasets.** Accuracy of embeddings and FUNGI features in kNN classification over 11 datasets, for two AugReg (Dosovitskiy et al., 2021; Steiner et al., 2022) ViT-B/16 models from timm (Wightman, 2019) pretrained on IN1K and IN21K.

| | Pretrain | Cars | CUB | DTD | ESAT | C100 | C10 | Pets | Food | IN1K | FGVC | Flowers | Mean |
|---|---|---|---|---|---|---|---|---|---|---|---|---|---|
| **Full dataset** | | | | | | | | | | | | | |
| Embeddings | IN1K | 21.3 | 42.0 | 54.3 | 89.0 | 66.3 | 89.4 | 87.3 | 52.3 | 77.2 | 17.9 | 53.8 | 59.2 |
| FUNGI | IN1K | 27.2 | 50.1 | 58.6 | 93.4 | 69.7 | 90.7 | 89.5 | 58.9 | 78.8 | 21.4 | 61.6 | 63.6 ↑4.4 |
| Embeddings | IN21K | 21.0 | 74.0 | 58.4 | 91.8 | 58.4 | 82.9 | 83.6 | 70.6 | 72.1 | 23.0 | 95.0 | 66.4 |
| FUNGI | IN21K | 25.1 | 74.2 | 65.0 | 94.7 | 63.5 | 85.7 | 85.7 | 73.4 | 74.5 | 24.3 | 96.6 | 69.3 ↑2.9 |
| **5-Shot** | | | | | | | | | | | | | |
| Embeddings | IN1K | 9.4 | 23.7 | 32.5 | 38.6 | 36.9 | 48.8 | 57.5 | 20.1 | 55.7 | 8.3 | 41.2 | 33.9 |
| FUNGI | IN1K | 11.4 | 26.6 | 33.9 | 42.2 | 38.6 | 50.2 | 59.4 | 24.1 | 58.6 | 9.2 | 49.8 | 36.7 ↑2.8 |
| Embeddings | IN21K | 7.6 | 50.0 | 33.7 | 47.7 | 23.2 | 39.7 | 53.3 | 32.0 | 40.3 | 10.7 | 86.2 | 38.6 |
| FUNGI | IN21K | 9.2 | 48.5 | 36.3 | 54.5 | 28.2 | 41.7 | 51.0 | 37.8 | 45.4 | 12.2 | 85.8 | 41.0 ↑2.4 |

label using an attention mechanism. Images are first resized to $512 \times 512$, and then encoded as a set of $32^2 = 1024$ patch features using a ViT with patch size 16.

We enhance the patch features using SimCLR gradients, obtained by contrasting the input patch tokens against their nearest neighbors from a support index built with ScaNN (Guo et al., 2020). We use the reproduction of this evaluation protocol by Pariza et al. (2024) to run our experiments.

## 5 Experiments

In this section, we evaluate the performance of FUNGI in k-nearest neighbor image, text and audio classification and retrieval-based in-context scene understanding. Further experiments, including image retrieval, clustering and linear probing, are provided in Appendix B.

### 5.1 Image Classification

Following Caron et al. (2021), we evaluate our FUNGI features using the task of kNN classification. To show the generalizability of our method, we evaluate our features across ViT backbones (Dosovitskiy et al., 2021) with varying model sizes and pretraining strategies, including both supervised and self-supervised methods.

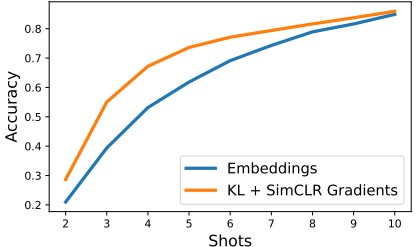

Figure 5: **Better data-efficiency.** kNN accuracy of embeddings and FUNGI (using only KL and SimCLR gradients) on ImageNet-100 using a DeIT-B/16 backbone when only $k$ shots are used.

We conduct our experiments on 11 diverse downstream datasets, described in Appendix D. Unless otherwise specified, we report the average accuracy across these datasets. We evaluate our features using the kNN implementation of scikit-learn (Pedregosa

Table 2: **Performance improves as more gradients are used.** Accuracy in image classification using kNN with embeddings and FUNGI features, averaged across 11 datasets for 7 backbones, for standard and few shot setups. Results for additional backbones are shown in Table 8. "K", "D" and "S" stand for KL, DINO and SimCLR, respectively.

|  |  | DINOv2 ViT-B/14 | DINO ViT-B/16 | DeIT ViT-B/16 | MoCov3 ViT-B/16 | DeIT ViT-B/32 | AugReg IN1K ViT-S/16 | AugReg IN1K ViT-B/16 |
|---|---|---|---|---|---|---|---|---|
| *full* | Embeddings | 79.9 | 69.0 | 65.3 | 63.2 | 61.7 | 60.8 | 59.2 |
|  | + K | 80.6 ↑0.7 | 69.4 ↑0.4 | 66.3 ↑1.0 | 63.4 ↑0.2 | 63.3 ↑1.6 | 60.3 ↓0.5 | 58.9 ↓0.3 |
|  | + K + D | 81.3 ↑1.4 | 70.1 ↑1.1 | 68.1 ↑2.8 | 64.7 ↑1.5 | 65.7 ↑4.0 | 62.6 ↑1.8 | 61.1 ↑1.9 |
|  | + K + D + S | **81.7** ↑1.8 | **70.9** ↑1.9 | **70.1** ↑4.8 | **65.8** ↑2.6 | **67.3** ↑5.6 | **64.3** ↑3.5 | **63.6** ↑4.4 |
| *few shot* | Embeddings | 47.6 | 39.3 | 38.4 | 34.7 | 36.2 | 34.9 | 33.9 |
|  | + K | 48.1 ↑0.5 | 39.4 ↑0.1 | 38.7 ↑0.3 | 35.8 ↑1.1 | 36.6 ↑0.4 | 34.2 ↓0.7 | 33.5 ↓0.4 |
|  | + K + D | 49.1 ↑1.5 | 39.7 ↑0.4 | 39.1 ↑0.7 | 36.6 ↑1.9 | 37.6 ↑1.4 | 35.0 ↑0.1 | 34.5 ↑0.6 |
|  | + K + D + S | **50.3** ↑2.7 | **40.5** ↑1.2 | **41.1** ↑2.7 | **38.2** ↑3.5 | **39.0** ↑2.8 | **36.5** ↑1.6 | **36.7** ↑2.8 |

Table 3: **FUNGI features improve in-context semantic segmentation.** mIoU for retrieval-based semantic segmentation on Pascal VOC 2012, comparing a DINO baseline against FUNGI features and the self-supervised HummingBird model. Results from Balazevic et al. (2024) are marked with ‡. We resize each image to $512 \times 512$ and extract $32^2 = 1024$ patch features.

|  |  | Memory Bank Size | | |
|---|---|---|---|---|
| Backbone | Features | $1024 \times 10^2$ | $1024 \times 10^3$ | $1024 \times 10^4$ |
| DINO ViT-S/16 | Embeddings | 37.2 | 43.1 | 46.6 |
| DINO ViT-S/16 | FUNGI | **50.7** ↑13.5 | **56.3** ↑13.2 | **58.0** ↑11.4 |
| DINO ViT-B/16 | Embeddings | 44.9 | 50.8 | 55.7 |
| DINO ViT-B/16 | FUNGI | **62.1** ↑17.2 | **66.1** ↑15.3 | **67.0** ↑11.3 |
| HummingBird ViT-B/16[‡] | Embeddings | - | - | **70.5** |

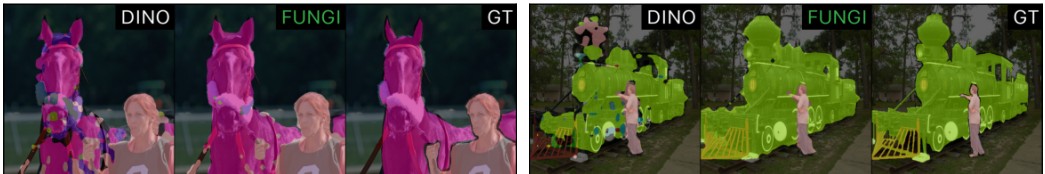

Figure 7: **FUNGI produces sharper and more complete segmentation masks.** Segmentation masks produced via nearest neighbor retrieval using DINO features (left), FUNGI (center) and the ground truth (right). Both methods use a memory bank of $1024 \times 10^4$ patches.

et al., 2011) with majority voting over 20 neighbors, for full dataset and few shot scenarios, the latter using five examples per class, to analyze the efficacy of our approach in low-data scenarios.

Our results, presented in Figure 6, show that FUNGI consistently improves the kNN performance of all ViT models, regardless of model size or pretraining strategy, both for the full dataset and in few shot scenarios. We further investigate data-efficient settings in Figure 5, where FUNGI shows a significant improvement when 3 to 6 shots are used, highlighting the potential of FUNGI in low-data regimes.

In Table 1, we show that, with some exceptions, FUNGI provides consistent improvements across datasets for two AugReg (Steiner et al., 2022) ViT-B/16 backbones, pretrained on IN1K and IN21K, with FUNGI providing better results on the former. We further discuss these results in Section 6.

Lastly, in Table 2 we show that performance improves as more gradients from different self-supervised objectives are used.

## 5.2 In-Context Scene Understanding

In this section, we assess the effectiveness of our approach in the task of retrieval-based semantic segmentation on Pascal VOC 2012 (Everingham et al., 2010) and ADE20K (Zhou et al., 2019,

Table 4: **Data-efficient semantic segmentation.** mIoU scores for data-efficient retrieval-based semantic segmentation on Pascal VOC 2012 and ADE20K, using DINO backbones and their FUNGI features and embeddings. We also compare FUNGI to end-to-end fine-tuning and find our method to perform best for VOC. Results from Balazevic et al. (2024) are marked with ‡.

| | | | Dataset Size | | | |
|---|---|---|---|---|---|---|
| | | | Pascal VOC | | ADE20K | |
| Backbone | Features | Decoder | 1/128 ($n = 83$) | 1/64 ($n = 165$) | 1/128 ($n = 158$) | 1/64 ($n = 316$) |
| ViT-B/16[‡] | - | E2E FT | 36.1 | 44.3 | **11.7** | **14.4** |
| ViT-S/16 | Emb. | NN | 26.3 | 31.8 | 8.8 | 10.0 |
| ViT-S/16 | FUNGI | NN | **29.1** ↑2.8 | **34.0** ↑2.2 | **10.2** ↑1.4 | **12.3** ↑2.3 |
| ViT-B/16 | Emb. | NN | 32.2 | 39.0 | 9.3 | 11.3 |
| ViT-B/16 | FUNGI | NN | **38.0** ↑5.8 | **46.8** ↑7.8 | **11.7** ↑2.4 | **13.7** ↑2.4 |

Table 5: **FUNGI features are useful for the text modality.** Top-1 accuracy in kNN text classification for the full dataset and few shot setups. "K" and "S" stand for KL and SimCLR, respectively.

| | TREC | | Banking-77 | | SST (Fine Grained) | | AG News | | Tweet-Eval | |
|---|---|---|---|---|---|---|---|---|---|---|
| | Full | 5-shot | Full | 5-shot | Full | 5-shot | Full | 10-shot | Full | 5-shot |
| **BERT Base** | | | | | | | | | | |
| Embeddings | 83.6 | 20.0 | 55.4 | 14.5 | 40.0 | 20.4 | 88.8 | 45.8 | 23.8 | 13.6 |
| + K | 85.6 ↑2.0 | 27.6 ↑7.6 | 67.1 ↑11.7 | 22.2 ↑7.7 | 40.7 ↑0.7 | 23.2 ↑2.8 | 91.0 ↑2.2 | 61.4 ↑15.6 | 24.4 ↑0.6 | 13.8 ↑0.2 |
| + K + S | **86.8** ↑3.2 | 23.0 ↑3.0 | **67.9** ↑12.5 | **23.8** ↑9.3 | **41.8** ↑1.8 | 18.4 ↓2.0 | 89.6 ↑0.8 | **61.9** ↑16.1 | **24.8** ↑1.0 | **14.5** ↑0.9 |
| **T5 Small** | | | | | | | | | | |
| Embeddings | **88.6** | **25.6** | 29.7 | 5.2 | 30.0 | **25.9** | 71.8 | 37.4 | 23.4 | 8.4 |
| + K | **88.6** | 23.6 ↓2.0 | **33.3** ↑3.6 | 5.6 ↑0.4 | **32.7** ↑2.7 | 24.1 ↓1.8 | 74.3 ↑2.5 | 40.6 ↑3.2 | 24.2 ↑0.8 | 9.5 ↑1.1 |
| + K + S | 88.4 ↓0.2 | 23.6 ↓2.0 | 29.1 ↓0.6 | **6.1** ↑0.9 | 32.0 ↑2.0 | 24.2 ↓1.7 | **74.8** ↑3.0 | **41.0** ↑3.6 | **24.4** ↑1.0 | **9.9** ↑1.5 |

2017). We use the trainaug and train splits to build the memory banks for Pascal VOC and ADE20K, respectively, and report the mean intersection over union (mIoU) on the validation set.

We apply FUNGI to DINO ViT-S/16 and ViT-B/16 models. Our results, presented in Table 3 and Table 7, demonstrate that FUNGI significantly enhances DINO's performance across all memory bank sizes, with the most substantial improvements observed in smaller memory banks for Pascal VOC. Qualitatively, FUNGI produces sharper and more complete segmentation masks, as shown in Figure 7. Notably, the DINO ViT-B/16 model, when enhanced with our FUNGI approach, achieves competitive results against the current state-of-the-art HummingBird model (Balazevic et al., 2024), with a difference of only 3.5% on Pascal VOC and 3.1% on ADE20K, **without** any training. This is a particularly promising result, as HummingBird employs a self-supervised pretraining strategy that is specialized for retrieval-based dense prediction tasks, which are the focus of our evaluation in this study.

In addition, we evaluate the efficacy of FUNGI in a data-efficient setup, and report the results in Table 4. Our findings indicate that our method outperforms DINO in this scenario, even when compared to end-to-end fine-tuning of DINO on the downstream task for Pascal VOC.

### 5.3 Other Modalities

**Natural language.** We evaluate FUNGI in the text domain using five datasets, described in Appendix D, and two transformer language models: BERT base uncased (Devlin et al., 2019) and T5-small (Raffel et al., 2020). We use the $\mathcal{L}_{KL}$ and $\mathcal{L}_{SimCLR}$ losses and obtain the SimCLR views by randomly deleting words with a 10% probability. The results are presented in Table 5 and show that FUNGI achieves improvements in the text domain. However, SimCLR gradients struggle with some datasets. Different data augmentation strategies, such as back-translation (Sennrich et al., 2016), or language-specific self-supervised losses, *e.g.*, masked language modeling (Devlin et al., 2019), may yield more discriminative gradients. We leave this investigation for future work. Furthermore, in Appendix B.5, we investigate the potential of FUNGI in language in-context learning.

**Audio.** We demonstrate gains for the audio modality in Table 12, where we improve the ESC-50 kNN classification accuracy from 42.8% to 47.0% and SpeechCommands from 27.4% to 29.9% with an SSAST backbone (Gong et al., 2022). Further details are provided in Appendix B.2.

Table 6: **Impact of the projection head configuration.** Top-1 accuracy of gradients on ImageNet-100 in k-nearest neighbor classification versus the projection head configuration for KL, DINO and SimCLR gradients. "norm" indicates whether the features are $L_2$-normalized before being projected. As features are always $L_2$-normalized for the SimCLR objective, the "empty" head configuration is not applicable. The default setup is marked in `cyan`.

| $\nabla_{KL}$ | | | | $\nabla_{DINO}$ | | | | $\nabla_{SimCLR}$ | | |
| --- | --- | --- | --- | --- | --- | --- | --- | --- | --- | --- |
| Norm | Projection | Acc. | | Norm | Projection | Acc. | | Norm | Projection | Acc. |
| | | 88.3 | | | | 79.3 | | | | N/A |
| ✓ | | 87.3 | | ✓ | | 87.8 | | ✓ | | 88.7 |
| | ✓ | 88.8 | | | ✓ | 84.7 | | | ✓ | **88.8** |
| ✓ | ✓ | **89.1** | | ✓ | ✓ | **90.1** | | ✓ | ✓ | 88.7 |

## 5.4 Ablation Studies

**Projection head.** To compute our self-supervised losses, we first $L_2$-normalize the model embeddings (except for SimCLR) and then project them using a randomly initialized linear head. We motivate this choice empirically by ablating these components, and the results in Table 6 show that this configuration produces the most predictive gradients for ImageNet-100.

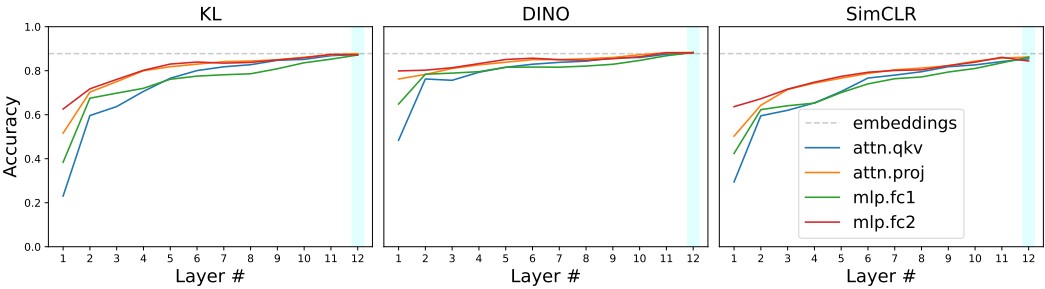

Figure 8: **Gradients from deeper layers are more predictive.** Top-1 accuracy of gradients obtained from every layer of a supervised DeIT ViT-B/16 in k-nearest neighbor classification on ImageNet-100 for the KL, DINO, and SimCLR objectives. The default setup (last layers) is marked in `cyan`.
.

**Gradients source layer.** Throughout the paper, we extract gradients from the self-attention output projection of the last transformer block. Intuitively, deeper layers provide more predictive features, and thus, their gradient should display the same behavior. This assumption is confirmed by our results in Figure 8, where, for all losses, deeper layers consistently produce more predictive gradients. Regarding the choice of layer within a transformer block, for shallower blocks, the second MLP layer is significantly more predictive, but the performance gap becomes insignificant as we move towards deeper blocks, favoring (by a small margin) the attention output projection, which is also more memory efficient, as it has fewer parameters compared to other layers.

## 6 Discussion and Conclusion

**Broader impact.** Our method improves the features used for the kNN algorithm. As such, it is a fundamental contribution to Machine Learning. Given the ubiquitous use of kNN, our method could have positive consequences, such as improving reliability and factuality in Retrieval Augmented Generation (RAG) systems, where Language Models are grounded in retrieved pieces of text before generating an answer. We do not foresee any direct negative consequence caused by our method.

**Impact of pretraining dataset.** Our method works with various backbones, model sizes, and pretraining strategies. However, we have observed that the benefits diminish as the size of the pretraining dataset increases: in Table 1, FUNGI provides a smaller relative improvements on a backbone pretrained with IN21K compared to one pretrained on IN1K, and similarly, in Table 16 the relative improvement over EVA-CLIP (Sun et al., 2023) is smaller compared to the improvement over CLIP (Radford et al., 2021), as they are pretrained on 2B and 400M text-image pairs respectively.

**Computational efficiency.** Computing FUNGI features introduces an overhead, which we measure in Table 27 by comparing the throughput of a DeIT ViT-B/16 when extracting gradients and embeddings. The DINO and SimCLR losses have the largest overhead, as they forward 12 and 49 views per image, respectively. As shown in Appendix C.1, this number can be reduced, at a performance cost. However, thanks to our dimensionality reduction, the speed of kNN retrieval is not impacted.

## 7 Conclusion

We have shown that gradients from self-supervised objectives have predictive abilities and encode complementary information to the model embeddings. Building on those findings, we introduced FUNGI , which effectively combines embeddings and gradients into powerful features for retrieval-based tasks. Specifically, we have shown that FUNGI enhance the performance of kNN-based image and text classification across models, pretraining strategies, and downstream datasets, both for full dataset and few shot setups. Moreover, we have shown that FUNGI significantly boost the performance of DINO features for retrieval-based semantic segmentation tasks.

**Acknowledgements.** We acknowledge the use of the Dutch national supercomputer Snellius to run the experiments presented in this paper. YMA thanks Tengda Han for the initial discussions on using self-supervised gradients for tasks other than learning. WS thanks the Amsterdam ELLIS Unit for their generous funding, which allowed him to visit the Valeo laboratory in Paris. The presentation of this paper at the conference was financially supported by the Amsterdam ELLIS Unit and Qualcomm.

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

# Appendix

## Table of Contents

## A  Algorithm

Algorithm 1 provides pytorch-style pseudocode for the computation of $\mathcal{L}_{\mathrm{KL}}$, the gradient extraction, and the computation of FUNGI features (without PCA).

---

**Algorithm 1** PyTorch pseudocode for the KL FUNGI features.

---

```
# model, head, proj: the vision backbone, head and the random projection used to downsample gradients
proj = (torch.rand(feat_dim, grad_dim) - 0.5) > 0
uniform = torch.ones(feat_dim) / feat_dim

for x in dataset:
    # Extract the feature and its projection
    y = model(x)
    z = head(y)

    kl_div(log_softmax(z), softmax(uniform)).backward() # Calculate the loss and backpropagate
    layer = model.blocks.11.attn.proj # Select the target layer

    # Extract and project the gradients
    gradients = torch.cat([layer.weight.grad, layer.bias.grad.unsqueeze(dim=-1)], dim=-1).view(-1)
    gradients = proj @ gradients.view(-1)

    feature = torch.cat([normalize(y), normalize(gradients)], dim=-1) # Build the final feature
```

---

## B  Additional Experimental Results

In this section we first illustrate additional results that solidify the findings discussed in the main paper, including the evaluation of FUNGI in image retrieval, k-nearest neighbor audio classification, linear classification and clustering and a brief investigation of the performance of gradients from clustering and masked image modeling self-supervised objectives.

In Table 8, we report the performance of embeddings and FUNGI features for some additional backbones, including CLIP and EVA-CLIP models, for which, as explained in the main text, we experience diminishing returns as the pretraining dataset size grows. Moreover, for both models, the SimCLR loss does not produce predictive gradients, which we hypothesize is due to models being saturated to a contrastive loss, as they use a similar objective for pretraining.

In Table 10, we report the mean accuracy and one standard deviation (computed via `numpy.std`) across three seeds for a subset of our datasets, using a DeIT ViT-B/16 backbone, showing that the performance improvements of FUNGI are consistent across seeds.

In Figure 11 we evaluate the effectiveness of FUNGI across different ViT model sizes. The findings show that FUNGI improves the results for all three ViT models (ViT-S, ViT-B, and ViT-L), with the most significant improvements observed in the ViT-B model. In Table 9, we provide further evidence for the scalability of FUNGI by evaluating it on several ViT-L and ViT-H backbones.

In Table 7, we report the performance of FUNGI for in-context retrieval-based semantic segmentation on ADE20K, and show that our method outperforms DINO across all memory bank sizes and is competitive against HummingBird.

Table 7: **FUNGI features improve in-context semantic segmentation on ADE20K.** We report the mIoU for retrieval-based semantic segmentation on ADE20K, comparing a DINO baseline against FUNGI features and the self-supervised HummingBird model. Results from Balazevic et al. (2024) are marked with ‡. We resize each image to $512 \times 512$ and extract $32^2 = 1024$ patch features.

| Backbone | Features | Memory Bank Size | | |
| --- | --- | --- | --- | --- |
| | | $1024 \times 10^2$ | $1024 \times 10^3$ | $1024 \times 10^4$ |
| DINO ViT-S/16 | Embeddings | 11.4 | 14.5 | 17.0 |
| DINO ViT-S/16 | FUNGI | **16.1** ↑4.7 | **20.0** ↑5.5 | **22.3** ↑5.3 |
| DINO ViT-B/16 | Embeddings | 14.5 | 18.3 | 20.8 |
| DINO ViT-B/16 | FUNGI | **19.2** ↑4.7 | **23.5** ↑5.2 | **25.2** ↑4.4 |
| HummingBird ViT-B/16‡ | Embeddings | - | - | **28.3** |

Table 8: **Additional backbones.** Average accuracy of embeddings and FUNGI features in k-nearest neighbor classification across 11 datasets for CLIP (Radford et al., 2021; Sun et al., 2023), AugReg (Steiner et al., 2022), DeIT III (Touvron et al., 2022) and masked autoencoder (He et al., 2022) models. "K", "D" and "S" stand for KL, DINO and SimCLR, respectively.

| | EVA CLIP ViT-B/16 | CLIP ViT-B/16 | AugReg IN21K ViT-B/16 | AugReg IN21K ViT-S/16 | DeIT III ViT-B/16 | MAE ViT-B/16 |
| --- | --- | --- | --- | --- | --- | --- |
| **Full Dataset** | | | | | | |
| Embeddings | 83.2 | 73.7 | 66.4 | 65.6 | 64.2 | 24.0 |
| + K | 83.8 ↑0.6 | 76.9 ↑3.2 | 67.1 ↑0.7 | 65.3 ↓0.3 | 64.8 ↑0.6 | 44.4 ↑20.4 |
| + K + D | **84.1** ↑0.9 | **77.7** ↑4.0 | 68.6 ↑2.2 | 67.1 ↑1.5 | 67.3 ↑3.1 | **45.4** ↑21.4 |
| + K + D + S | 83.4 ↑0.2 | 74.6 ↑0.9 | **69.3** ↑2.9 | **67.6** ↑2.0 | **68.2** ↑4.0 | 38.8 ↑14.8 |
| **Few Shot** | | | | | | |
| Embeddings | 53.1 | 43.0 | 38.6 | 37.4 | 34.9 | 7.5 |
| + K | 54.0 ↑0.9 | 47.2 ↑4.2 | 40.2 ↑1.6 | 37.7 ↑0.3 | 36.2 ↑1.3 | 18.5 ↑11.0 |
| + K + D | **54.1** ↑1.0 | **47.9** ↑4.9 | 40.3 ↑1.7 | 38.7 ↑1.3 | 37.2 ↑2.3 | **19.2** ↑11.7 |
| + K + D + S | 53.7 ↑0.6 | 44.4 ↑1.4 | **41.0** ↑2.4 | **39.5** ↑2.1 | **39.6** ↑4.7 | 14.0 ↑6.5 |

## B.1 Image Retrieval

We evaluate the performance of FUNGI features in image retrieval using the revisited (Radenović et al., 2018) Oxford (Philbin et al., 2007) and Paris (Philbin et al., 2008) landmarks datasets. We report the mean average precision (mAP) for both the medium (M) and hard (H) splits.

For this task, we use FUNGI features built with DINO and KL gradients, as the SimCLR gradients did not result in good retrieval performance. The results displayed in Table 11 show that FUNGI features improve the retrieval abilities of all backbones, except for DINOv2. Our method is particularly effective when applied on CLIP backbones: on the Paris hard split, we improve by 12.4% and 7.2% for CLIP and EVA-CLIP, respectively.

Table 9: **Scalability experiments.** Average accuracy of embeddings and FUNGI features in k-nearest neighbor classification across 11 datasets (7 for ViT-H) for ViT-L and H backbones. "K", "D" and "S" stand for KL, DINO and SimCLR, respectively.

| | DINOv2 ViT-L/16 | CLIP ViT-L/14 | DeIT ViT-H/14 | AugReg IN21K ViT-L/16 |
|---|---|---|---|---|
| **Full Dataset** | | | | |
| Embeddings | 80.5 | 80.2 | 77.2 | 74.7 |
| + K | 81.2 ↑0.7 | 82.4 ↑2.2 | 77.3 ↑0.1 | 75.0 ↑0.3 |
| + K + D | 81.6 ↑1.1 | **82.9** ↑2.7 | 78.0 ↑0.8 | 76.2 ↑1.5 |
| + K + D + S | **82.3** ↑1.8 | 81.1 ↑0.9 | **78.8** ↑1.6 | **76.5** ↑1.8 |
| **Few Shot** | | | | |
| Embeddings | 47.1 | 48.5 | 48.3 | 47.7 |
| + K | 48.6 ↑1.5 | 51.8 ↑3.3 | 47.7 ↓0.6 | 48.3 ↑0.6 |
| + K + D | 49.0 ↑1.9 | **52.9** ↑4.4 | 46.9 ↓1.4 | 48.5 ↑0.8 |
| + K + D + S | **51.4** ↑4.3 | 50.3 ↑1.8 | **50.4** ↑2.1 | **50.2** ↑2.5 |

Table 10: **FUNGI is consistent across seeds.** Average accuracy in kNN classification and one standard deviation for FUNGI features on 8 datasets, measured across three seeds using a DeIT ViT-B/16 backbone. "K", "D" and "S" stand for KL, DINO and SimCLR, respectively.

| | Cars | CUB | DTD | ESAT | Pets | Food | FGVC | Flowers |
|---|---|---|---|---|---|---|---|---|
| Embeddings | 32.3 | 56.0 | 58.6 | 90.7 | 90.8 | 60.5 | 23.5 | 56.9 |
| + K | $33.5 \pm 0.2$ | $57.9 \pm 0.1$ | $60.4 \pm 0.2$ | $91.6 \pm 0.2$ | $91.3 \pm 0.2$ | $61.5 \pm 0.1$ | $22.9 \pm 0.1$ | $60.7 \pm 0.6$ |
| + K + D | $36.2 \pm 0.2$ | $60.9 \pm 0.2$ | $63.1 \pm 0.3$ | $93.4 \pm 0.1$ | $91.8 \pm 0.0$ | $65.2 \pm 0.2$ | $24.2 \pm 0.4$ | $64.3 \pm 0.5$ |
| + K + D + S | $\mathbf{39.3} \pm 0.3$ | $\mathbf{63.6} \pm 0.3$ | $\mathbf{64.1} \pm 0.2$ | $\mathbf{95.0} \pm 0.2$ | $\mathbf{92.1} \pm 0.2$ | $\mathbf{67.3} \pm 0.1$ | $\mathbf{28.3} \pm 0.6$ | $\mathbf{69.3} \pm 0.5$ |

## B.2 Audio Classification

We evaluate FUNGI on the audio modality using SSAST (Gong et al., 2021, 2022), a self-supervised audio spectrogram transformer trained for audio and speech classification, as the backbone. We construct FUNGI features using KL and SimCLR gradients, and test their performance in k-nearest neighbor classification on ESC 50 (Piczak, 2015) and SpeechCommands V2 (Warden, 2018).

We use the same formulation as in the vision experiments for the $\mathcal{L}_{KL}$ and $\mathcal{L}_{SimCLR}$ objectives. However, for the latter, we obtain 16 views of the same clip by adding uniform noise following Gong et al. (2022). If we define the filter bank of an audio clip as $c \in \mathbb{R}^{h,w}$, the noise-augmented clip $\hat{c}$ is computed as:

$$\hat{c} = c + \frac{x_1 \cdot x_2}{10} \qquad x_1 \sim U(0,1)^{h,w} \qquad x_2 \sim U(0,1). \tag{7}$$

Finally, $\hat{c}$ is shifted by a factor sampled from a discrete uniform distribution $U(-10, 10)$. The complete list of hyperparameters used for the audio classification experiments is reported in Table 13.

The results are reported in Table 12, and show that FUNGI features built using KL gradients yield promising results, improving by up to 4.2% on the baseline. On the other hand, using SimCLR gradients does not consistently yield improvements. It rather often causes a performance drop when combined with KL gradients. As with text classification, further research is needed to determine the optimal self-supervised objectives and data augmentation to extract predictive gradients.

## B.3 Linear Classification of Image Features

We evaluate FUNGI features in logistic regression, using the implementation from the cyanure library (Mairal, 2019). We train each classifier for a maximum of 300 epochs (30 in the case of ImageNet-1K) using $L_2$ regularization. For each dataset and feature combination (i.e., embeddings, embeddings + $\nabla_{KL}$, etc.), we pick the best regularization parameter between 5 linearly spaced values in the interval $[5 \times 10^{-6}, 5 \times 10^{-4}]$ using the validation set. For datasets without a validation set, we generate one using an 80/20 stratified split. The final model is trained using the entire training dataset.

We report the results in Table 15 and Table 16 and find that, in linear classification, FUNGI features are most effective for backbones pretrained using a supervised objective. In contrast, self-supervised backbones do not benefit as much. This is especially evident for DINO and DINOv2, where

Table 11: **FUNGI improves image retrieval**. Mean average precision (mAP) of embeddings and FUNGI for retrieval on the Paris (Philbin et al., 2008) and Oxford (Philbin et al., 2007) landmarks datasets, for both medium (M) and hard (H) splits. "K" and "D" stand for KL and DINO, respectively.

| | DINOv2 ViT-B/14 | | DINO ViT-B/16 | | CLIP ViT-B/16 | | EVA CLIP ViT-B/16 | | DeIT ViT-B/16 | |
|---|---|---|---|---|---|---|---|---|---|---|
| **Oxford** | M | H | M | H | M | H | M | H | M | H |
| Embeddings | 69.7 | 42.0 | 39.2 | 11.0 | 31.4 | 10.8 | 36.7 | 12.7 | 36.6 | 12.6 |
| + K | **70.4** ↑0.7 | **42.6** ↑0.6 | 38.6 ↓0.6 | 11.6 ↑0.6 | 40.4 ↑9.0 | 14.5 ↑3.7 | 39.9 ↑3.2 | 13.9 ↑1.2 | 36.9 ↑0.3 | 12.6 ↑0.0 |
| + D | 69.4 ↓0.3 | 41.2 ↓0.8 | **40.4** ↑1.2 | 12.9 ↑1.9 | 41.4 ↑10.0 | **15.3** ↑4.5 | 40.9 ↑4.2 | 14.8 ↑2.1 | 38.8 ↑2.2 | 12.7 ↑0.1 |
| + K + D | 70.1 ↑0.4 | 42.0 ↑0.0 | 39.8 ↑0.6 | **13.0** ↑2.0 | **42.6** ↑11.2 | 14.7 ↑3.9 | **41.2** ↑4.5 | **14.9** ↑2.2 | **39.1** ↑2.5 | **12.8** ↑0.2 |
| **Paris** | M | H | M | H | M | H | M | H | M | H |
| Embeddings | **89.4** | **77.5** | 63.8 | 37.6 | 64.6 | 40.4 | 69.8 | 46.7 | 63.0 | 37.2 |
| + K | 88.7 ↓0.7 | 76.2 ↓1.3 | 64.5 ↑0.7 | 38.4 ↑0.8 | 74.7 ↑10.1 | **52.8** ↑12.4 | **75.2** ↑5.4 | **53.9** ↑7.2 | 63.6 ↑0.6 | 37.7 ↑0.5 |
| + D | 89.0 ↓0.4 | 76.2 ↓1.3 | 65.6 ↑1.8 | 39.0 ↑1.4 | 72.0 ↑7.4 | 47.8 ↑7.4 | 72.9 ↑3.1 | 50.2 ↑3.5 | **65.7** ↑2.7 | **40.2** ↑3.0 |
| + K + D | 88.7 ↓0.7 | 75.9 ↓1.6 | **65.8** ↑2.0 | **39.5** ↑1.9 | **75.1** ↑10.5 | 52.5 ↑12.1 | 74.9 ↑5.1 | 53.0 ↑6.3 | 65.6 ↑2.6 | 40.0 ↑2.8 |

Table 12: **FUNGI works for audio.** Top-1 accuracies in k-nearest neighbor audio classification of embeddings and FUNGI features obtained from a SSAST backbone (Gong et al., 2022, 2021). "K" and "S" stand for KL and SimCLR, respectively.

| | ESC 50 | | SpeechCommands V2 | |
|---|---|---|---|---|
| | Full | 5-shot | Full | 5-shot |
| Embeddings | 42.8 | 20.0 | 27.4 | 5.3 |
| + K | **47.0** ↑4.2 | **21.2** ↑1.2 | **29.9** ↑2.5 | **6.1** ↑0.8 |
| + S | 45.2 ↑2.5 | 19.0 ↓1.0 | 25.4 ↓2.0 | 5.5 ↑0.2 |
| + K + S | 45.8 ↑3.0 | 21.0 ↑1.0 | 27.3 ↓0.1 | 5.8 ↑0.5 |

Table 13: **Audio classification experimental details.** Parameters used to extract audio encoder gradients for the $\mathcal{L}_{\text{KL}}$ (left) and $\mathcal{L}_{\text{SimCLR}}$ (right) objectives.

| Parameter | Value |
|---|---|
| Temperature | 15 |
| Projection Dim | 768 |

| Parameter | Value |
|---|---|
| (Positive, Negative) Views | 16, 2 |
| Projection Dim | 768 |
| Negatives Batch Size | $64 \times 2$ |
| Temperature | 0.07 |

FUNGI often yields worse results, especially in a few shot scenarios. Contrary to the k-nearest neighbor classification results, the best feature combination is backbone specific, and in Figure 9, we show that significant performance improvements can be achieved by picking the best feature combination for each backbone.

## B.4 Clustering

We evaluate the performance of FUNGI features built with KL and DINO gradients in k-means clustering (Lloyd, 1982) of image features using faiss (Johnson et al., 2019; Douze et al., 2024). Given a dataset with $C$ classes, we compute $C$ clusters via k-means and match clusters to classes using the Hungarian algorithm (Kuhn, 1955). We then measure the average overlap between clusters and classes. The results in Table 14 show that, on average, FUNGI features built with KL and DINO gradients improve the overlap between clusters and classes. In particular, FUNGI can improve the clustering performance by up to 15.8% on Oxford-IIT Pets for a CLIP ViT-L/14 backbone.

## B.5 Language In-Context Learning

Liu et al. (2022) has shown that selecting examples for in-context learning using retrieval outperforms a random baseline, and that using encoders fine-tuned on a task similar to the target one further improves performance. Thus, we hypothesize that using FUNGI features to retrieve in-context examples can improve performance, as they contain an adaptation signal to the task at hand. We test this hypothesis by measuring the in-context classification performance of GPT 4o mini (Achiam et al., 2023) using examples retrieved either using embeddings or FUNGI features built with KL and

Table 14: **FUNGI improves clustering.** Overlap between classes and clusters found via k-means clustering of embeddings and FUNGI features. "K" and "D" stand for KL and DINO respectively.

| | Cars | CUB | DTD | ESAT | C100 | C10 | Pets | Food | IN1K | FGVC | Flowers | Mean |
|---|---|---|---|---|---|---|---|---|---|---|---|---|
| **CLIP ViT-L/14** | | | | | | | | | | | | |
| Embeddings | 49.4 | 52.8 | 44.7 | 52.4 | 44.5 | 65.0 | 55.6 | **72.9** | 50.9 | 29.1 | **69.4** | 53.3 |
| + K | 55.0 | **57.9** | 53.1 | **55.9** | 56.1 | **68.5** | 67.5 | 71.0 | 55.3 | 33.8 | 66.8 | 58.3 ↑5.0 |
| + K + D | **56.2** | 57.0 | **55.4** | 55.4 | 55.9 | 66.5 | **71.4** | 72.7 | **55.7** | **35.1** | 66.6 | **58.9** ↑5.6 |
| **DINOv2 ViT-L/14** | | | | | | | | | | | | |
| Embeddings | 28.6 | 63.7 | **50.6** | 55.8 | 70.3 | 77.6 | **79.1** | 68.8 | 61.4 | 21.5 | 79.2 | 59.7 |
| + K | **29.3** | 64.0 | 48.5 | 55.9 | 67.2 | 84.5 | 74.2 | 67.9 | 59.9 | 20.3 | 80.2 | 59.2 ↓0.5 |
| + K + D | 28.9 | **64.9** | 48.3 | **66.6** | **70.5** | **85.2** | 75.2 | 66.5 | 60.3 | 22.0 | **81.2** | **60.8** ↑1.1 |
| **ViT ViT-B/16** | | | | | | | | | | | | |
| Embeddings | 15.8 | 33.5 | 43.9 | 55.7 | **50.0** | **83.9** | 76.0 | 28.9 | **78.5** | 14.9 | 41.3 | 47.5 |
| + K | **16.3** | 34.5 | 44.3 | **65.1** | 49.2 | 82.7 | **79.6** | 28.6 | 77.5 | 14.5 | 43.0 | 48.7 ↑1.2 |
| + K + D | 17.8 | **36.7** | **48.7** | 61.2 | 49.0 | 82.1 | 77.1 | **31.8** | 76.6 | **15.4** | 49.2 | **49.6** ↑2.1 |

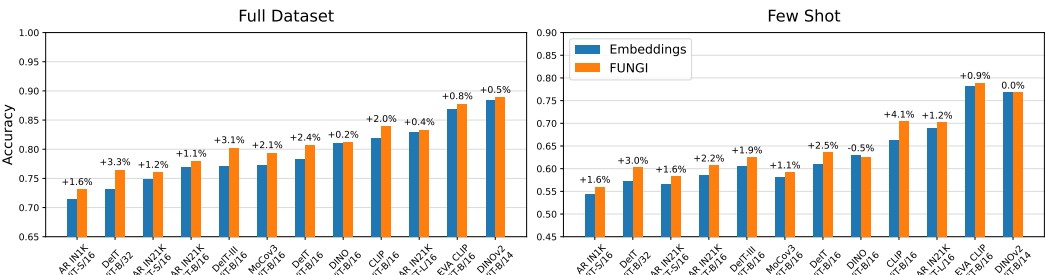

Figure 9: **FUNGI works across backbones for linear probing.** Accuracy in logistic regression-based image classification of embeddings and FUNGI features on various ViT backbones, both for full dataset and few shot setups, averaged over 11 datasets. For the FUNGI features, we chose the best performing combination across datasets. "AR" indicates AugReg backbones (Steiner et al., 2022).

SimCLR gradients using a BERT backbone. We retrieve the top 20 most similar training examples for each test sample and ask the model to predict its label using a prompt template similar to the one shown in Listing 1. For a fair evaluation, we set the model temperature to 0. The results listed in Table 17 show that examples retrieved using FUNGI features improve the in-context learning accuracy by up to 2.5% on banking-77.

```
You have to annotate banking-related queries with an appropriate intent.
You must choose a single class in the following comma-separated list:

{list of possible labels, comma-separated}

You must reply only with the class name, nothing more. Here's some examples:

{(text, label) pairs from the training set}

The query sample is: {query text}
```

Listing 1: The prompt used for the in-context learning evaluation of embeddings and FUNGI features on banking-77 using a GPT 4o mini backbone. The labels are given as strings, e.g. `exchange_rate`.

## B.6 Additional Self-Supervised Objectives

In this section, we study the performance of gradients obtained by two additional self-supervised objectives, DeepCluster (Caron et al., 2018) and iBOT (Zhou et al., 2022) in k-nearest neighbor classification on ImageNet-100 using a DeIT ViT-B/16 backbone. DeepCluster is a self-distillation and explicit clustering-based self-supervised method that alternates between clustering image features and training a model to predict cluster assignments. iBOT is an extension of DINO that combines image and patch-level objectives, the latter implemented as a latent-space masked image modeling (MIM) objective, where a learnable patch token replaces a subset of patches, and the model must reconstruct them.

Table 15: **The best gradients for linear probing are backbone-specific for the main backbones.** Average accuracy across 11 datasets for logistic regression-based image classification of embeddings and FUNGI features. "K", "D" and "S" stand for KL, DINO and SimCLR, respectively.

| | DINOv2 ViT-B/14 | DINO ViT-B/16 | DeIT ViT-B/16 | MoCov3 ViT-B/32 | DeIT ViT-B/32 | AugReg IN1K ViT-B/16 | AugReg IN1K ViT-S/16 |
|---|---|---|---|---|---|---|---|
| **Full Dataset** | | | | | | | |
| Embeddings | 88.3 | 81.0 | 78.2 | 77.3 | 73.1 | 71.6 | 71.4 |
| + K | 88.3 ↑0.0 | 80.4 ↓0.6 | 78.5 ↑0.3 | 77.7 ↑0.4 | 73.4 ↑0.3 | 70.9 ↓0.7 | 70.5 ↓0.9 |
| + K + D | **88.8** ↑0.5 | **81.2** ↑0.2 | **80.7** ↑2.5 | **79.4** ↑2.1 | 76.2 ↑3.1 | 73.0 ↑1.4 | **73.0** ↑1.6 |
| + K + D + S | 88.7 ↑0.4 | 80.7 ↓0.3 | 80.5 ↑2.3 | 78.7 ↑1.4 | **76.4** ↑3.3 | **73.1** ↑1.5 | **73.0** ↑1.6 |
| **Few Shot** | | | | | | | |
| Embeddings | **76.7** | **62.9** | 61.0 | 58.0 | 57.2 | 54.8 | 54.4 |
| + K | 76.3 ↓0.4 | 62.2 ↓0.7 | 61.7 ↑0.7 | 57.6 ↓0.4 | 57.8 ↑0.6 | 54.4 ↓0.4 | 53.5 ↓0.9 |
| + K + D | **76.7** ↑0.0 | 62.4 ↓0.5 | **63.5** ↑2.5 | **59.2** ↑1.2 | 60.2 ↑3.0 | 56.5 ↑1.7 | 55.9 ↑1.5 |
| + K + D + S | 76.6 ↓0.1 | 61.6 ↓1.3 | 63.4 ↑2.4 | 58.7 ↑0.7 | 60.1 ↑2.9 | **57.0** ↑2.2 | **56.0** ↑1.6 |

Table 16: **The best gradients for linear probing are backbone-specific for the additional backbones.** Average accuracy across 11 datasets for logistic regression-based image classification of embeddings and FUNGI features. "K", "D" and "S" stand for KL, DINO and SimCLR, respectively.

| | EVA CLIP ViT-B/16 | AugReg IN21K ViT-L/16 | CLIP ViT-B/16 | DeIT-III ViT-B/16 | AugReg IN21K ViT-B/16 | AugReg IN21K ViT-S/16 | MAE ViT-B/16 |
|---|---|---|---|---|---|---|---|
| **Full Dataset** | | | | | | | |
| Embeddings | 86.9 | 82.9 | 81.8 | 77.1 | 76.8 | 75.6 | 38.6 |
| + K | 87.2 ↑0.3 | 82.0 ↓0.9 | 82.6 ↑0.8 | 78.1 ↑1.0 | 76.1 ↓0.7 | 74.4 ↓1.2 | 63.4 ↑24.8 |
| + K + D | **87.8** ↑0.9 | **83.3** ↑0.4 | **83.9** ↑2.1 | **80.2** ↑3.1 | **77.9** ↑1.1 | 76.6 ↑1.0 | **66.2** ↑27.6 |
| + K + D + S | 87.7 ↑0.8 | 83.2 ↑0.3 | 83.0 ↑1.2 | 79.9 ↑2.8 | 77.8 ↑1.0 | **76.8** ↑1.2 | 63.1 ↑24.5 |
| **Few Shot** | | | | | | | |
| Embeddings | 78.0 | 68.9 | 66.2 | 60.5 | 58.4 | 57.6 | 23.9 |
| + K | 78.6 ↑0.6 | 69.0 ↑0.1 | 69.3 ↑3.1 | 60.3 ↓0.2 | 59.1 ↑0.7 | 57.6 ↑0.0 | 36.0 ↑12.1 |
| + K + D | **78.9** ↑0.9 | **70.1** ↑1.2 | **70.4** ↑4.2 | **62.5** ↑2.0 | **60.7** ↑2.3 | **59.3** ↑1.7 | **37.3** ↑13.4 |
| + K + D + S | 77.5 ↓0.5 | 69.5 ↑0.6 | 65.9 ↓0.3 | 61.7 ↑1.2 | 59.8 ↑1.4 | 58.6 ↑1.0 | 32.3 ↑8.4 |

The results are displayed in Figure 10, and show that the objectives used in this work achieve similar performances to the model embeddings, even surpassing them in the case of DINO. At the same time, iBOT and DeepCluster instead produce gradients with relatively poor predictive performance. For the former, a possible reason is that it incorporates a dense loss, whose gradients may not help to discriminate examples on the image level. Regarding DeepCluster, models pretrained using this strategy had worse performance in retrieval tasks compared to supervised pretraining (Caron et al., 2018), which may explain the poor retrieval abilities of its gradients.

## B.7 Additional Ablations

**DINO data augmentation and head.** To maximize the signal-to-noise ratio, we only use local and global crops for the DINO data augmentation. We validate this choice empirically, and the results in Table 18 show that random crops produce more discriminative gradients compared to the standard data augmentation policy. Moreover, we also empirically validate the choice of using two independent heads for the DINO loss in Table 18, showing that this choice is beneficial for kNN classification.

**Random Projections.** In order to reduce the dimensionality of the gradients we use random projections, an unsupervised dimensionality reduction technique based on the lemma by Lindenstrauss

Table 17: **FUNGI improves language in-context learning.** Classification accuracy of GPT 4o mini in language in-context learning with examples retrieved using embeddings or FUNGI features, both extracted from a BERT backbone.

| | Banking-77 | SST |
|---|---|---|
| Embeddings | 88.7 | 52.5 |
| + KL + SimCLR | **91.2** ↑2.5 | **52.9** ↑0.4 |

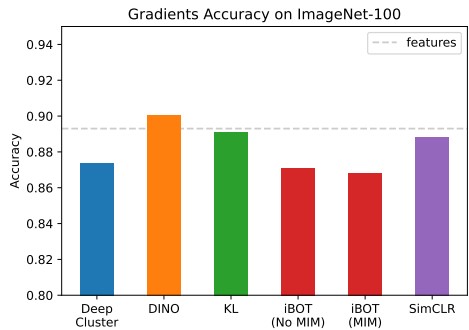
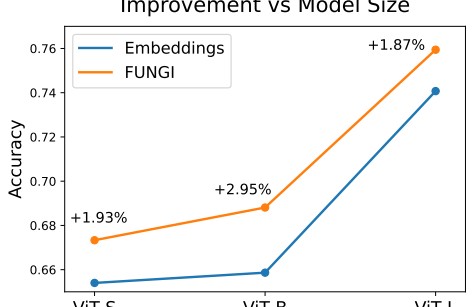

Figure 10: **Not all objectives produce good predictive gradients.** Top-1 accuracy in k-nearest neighbor classification of gradients obtained from different self-supervised objectives using a DeIT ViT-B/16 backbone. "MIM" stands for masked image modeling.

Figure 11: **Gains across backbone sizes.** Accuracy in k-nearest neighbor image classification averaged across 11 datasets using the model embeddings and FUNGI features extracted from AugReg backbones of increasing size.

Table 18: **DINO head configuration and data augmentation.** Top-1 accuracy on ImageNet-100 in k-nearest neighbor classification for the DINO gradients using shared or independent teacher and student heads (left) and with respect to the data augmentation policy (right).

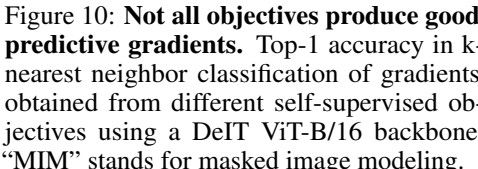

| Independent Heads | Accuracy | Data Augmentation | Accuracy |
|:---:|:---:|:---|:---:|
| ✗ | 88.4 | DINO | 88.9 |
| ✓ | **89.1** | Random Crops | **90.1** |

& Johnson (1984), which states that for a set of $N$ $d$-dimensional points and a projection subspace of dimension $k \geq k_0 = O(\epsilon^{-2}\log(N))$ there exist a mapping $f : \mathbb{R}^d \to \mathbb{R}^k$ that preserves euclidean distances with a distortion of at most $\epsilon \pm 1$. This mapping can be either implemented as a Gaussian, binary (Achlioptas, 2003) or sparse (Li et al., 2006) projection. Our method uses a binary projection, as it's more memory-efficient than a Gaussian matrix, and in Table 19, we compare its performance to the other initializations. The results show that the initialization has little impact on downstream performance, favoring binary projections by a small margin.

Table 19: **The random projection initialization has little impact on performance.** Comparison of the downstream accuracy of FUNGI features built with gradients projected using matrices with different initializations. We report the mean and one standard deviation measured across three runs using the Flowers102 dataset and a DeIT ViT-B/16 backbone. No further dimensionality reduction was applied to the concatenated features.

| | Binary | Gaussian | Sparse |
|:---|:---:|:---:|:---:|
| Embeddings | 56.9 | 56.9 | 56.9 |
| + K | $59.7 \pm 0.4$ | $\mathbf{60.2 \pm 0.2}$ | $60.0 \pm 0.1$ |
| + K + D | $\mathbf{63.5 \pm 0.1}$ | $63.2 \pm 0.5$ | $63.1 \pm 0.2$ |
| + K + D + S | $\mathbf{69.2 \pm 0.6}$ | $69.0 \pm 0.7$ | $68.9 \pm 0.9$ |

**PCA.** We use Principal Component Analysis (PCA) to combine data from multiple losses at an iso-storage and retrieval speed cost. Given a dataset of FUNGI features, we fit the PCA on the training split and use it to transform training and test splits. Table 20 lists the PCA dimensionalities used for each model architecture and shows that they do not cause a decrease in performance but rather provide a minor improvement on average. Moreover, we compare PCA against binary, Gaussian, and sparse random projections in Table 21 and find that all random projection-based methods result in a drop in accuracy compared to the original features, while PCA consistently improves performance.

Table 20: **PCA does not degrade performance.** PCA output dimensionalities with respect to the backbone architecture (left) and its impact on k-nearest neighbor image classification accuracy averaged across 11 datasets using a DeIT ViT-B/16 backbone (right).

| Architecture | PCA Dim | | No PCA | PCA |
|---|---|---|---|---|
| ViT-S/16 | 384 | Embeddings | 65.1 | **65.3** ↑0.2 |
| ViT-B/16, ViT-L/16 | 512 | + KL | 66.0 | **66.3** ↑0.3 |
| BERT, T5 | 512 | + KL + DINO | 67.8 | **68.1** ↑0.3 |
| SSAST | 512 | + KL + DINO + SimCLR | 69.8 | **70.1** ↑0.3 |

Table 21: **PCA is the best dimensionality reduction method.** The mean-per-class accuracy of embeddings and FUNGI features from a DeIT ViT-16/B backbone on Flowers102, transformed with PCA and random projections. We report the mean and one standard deviation across three seeds.

| | No Reduction | PCA | Proj. (Binary) | Proj. (Gaussian) | Proj. (Sparse) |
|---|---|---|---|---|---|
| Embeddings | $57.2 \pm 0.0$ | $\mathbf{61.6 \pm 0.0}$ | $55.5 \pm 0.4$ | $55.8 \pm 1.0$ | $56.0 \pm 0.5$ |
| + K | $59.4 \pm 0.0$ | $\mathbf{64.0 \pm 0.0}$ | $59.0 \pm 0.8$ | $58.2 \pm 0.8$ | $58.7 \pm 0.3$ |
| + K + D | $64.3 \pm 0.0$ | $\mathbf{68.3 \pm 0.0}$ | $62.9 \pm 0.4$ | $62.2 \pm 0.3$ | $61.9 \pm 0.5$ |
| + K + D + S | $69.2 \pm 0.0$ | $\mathbf{70.9 \pm 0.0}$ | $67.7 \pm 0.5$ | $67.0 \pm 0.4$ | $66.9 \pm 0.7$ |

# C  Experimental Details

## C.1  Vision Nearest Neighbor Classification Experimental Details

**Hyperparameters.** We use three losses to extract gradients from vision backbones: $\mathcal{L}_{\text{KL}}$, $\mathcal{L}_{\text{DINO}}$ and $\mathcal{L}_{\text{SimCLR}}$. The parameters used for each loss are shown in Table 22. This set of parameters is used consistently across backbones and datasets. While $\mathcal{L}_{\text{KL}}$ and $\mathcal{L}_{\text{DINO}}$ are robust to the choice of hyperparameters, $\mathcal{L}_{\text{SimCLR}}$ is particularly sensitive to the number of positive views, as shown in Figure 12, where performance increases in a logarithmic fashion as more positive views are used, at the cost of gradient extraction speed. While this behavior is consistent across datasets, it has the most significant impact in datasets with many classes, e.g., Flowers102.

**SimCLR data augmentation and loss details.** Given an image, we patchify it in 49 overlapping views as follows: we first resize the input image to $(224, 224)$, and then extract 49 patches of size $112 \times 112$, using a stride corresponding to $1/6$ of the patch size. No other style or color augmentation is used. As the number of patches increases, so does the memory required to compute the loss and the gradients. This problem can be partially tackled by precomputing the negative batch, which in our experiments is picked randomly from the training dataset and kept constant for every input. Moreover, we can observe that the SimCLR loss is only defined for positive pairs, so we only need to compute the similarity of positive pairs to all other pairs, significantly reducing the size of the similarities matrix and memory usage.

Table 22: **Image gradients setup.** Data augmentation and loss parameters used to extract gradients from vision encoders for $\mathcal{L}_{\text{KL}}$, $\mathcal{L}_{\text{SimCLR}}$ and $\mathcal{L}_{\text{DINO}}$ (left to right).

| Parameter | Value |
|---|---|
| Proj. Dim. | 768 |
| Temp. | 15 |

| Parameter | Value |
|---|---|
| Pos. Views | 49 |
| Neg. Views | 49 |
| Proj. Dim. | 96 |
| Neg. Batch | $256 \times 49$ |
| Temp. | 0.07 |

| Parameter | Value |
|---|---|
| Proj. Dim. | 2048 |
| Global Crops | 2 @ $224 \times 224$ |
| Global Crop Scale | 0.25, 1.0 |
| Local Crops | 10 @ $224 \times 224$ |
| Local Crop Scale | 0.05, 0.25 |
| Teacher Temp. | 0.07 |
| Student Temp. | 0.1 |

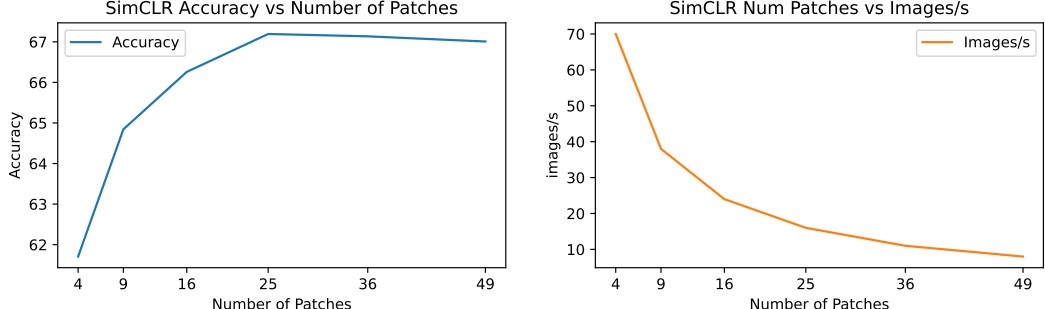

Figure 12: **SimCLR is sensitive to the number of views.** The SimCLR gradients mean-per-class accuracy on Flowers102 with respect to the number of patches (left) and the images/s versus the number of patches (right) using a supervised DeIT ViT-B/16 backbone.

## C.2 In-Context Scene Understanding Experimental Details

For the evaluation of the in-context scene understanding abilities of FUNGI features we closely replicate the setup described by Balazevic et al. (2024) for both the full and few shot setups, with two minor exceptions: (1) we use a single augmentation epoch for the full dataset evaluation and (2) we use an anisotropic quantization threshold of 0.2 for the nearest neighbor index, as this parameter was not specified in the paper. The full set of parameters for the evaluation, loss computation and data augmentation is reported in Table 23. As for data augmentation, we use the same policy of Balazevic et al. (2024), and apply each augmentation independently.

In order to construct FUNGI features for this task, we implement a SimCLR loss that contrasts patch tokens from an input image to their nearest neighbors retrieved from a supporting memory bank. In practice, we:

- Construct a memory bank of image patches of the same size as the one used for evaluation and its nearest neighbor index with ScaNN (Guo et al., 2020) following the procedure by Balazevic et al. (2024). We call this our support index.
- Then, for each image, we:
    1. Resize it to $512 \times 512$, compute its [CLS] and patch tokens and project them with a linear head. Excluding the [CLS] token, each image is mapped to $32^2 = 1024$ features, as all our backbones use patches of size 16.
    2. For each token, retrieve its two nearest neighbors from the support index and randomly drop 50% of them.
    3. Compute the SimCLR loss, where the patch tokens constitute the positive set and the neighbors the negative batch. This allows us to compute a per-patch gradient.
    4. Drop the [CLS] token, as it does not correspond to a real image patch.
    5. Construct FUNGI features as in Equation 8, where $f(x)$ maps an image to patches of dimension $d$, $L_2$ normalization is defined as $z' = z/||z||_2$ and cat indicates concatenation.

$$F = \mathtt{cat}'\left(\nabla'\mathcal{L}_{\mathrm{SimCLR}}, f'(x)\right) \qquad f(x): \mathbb{R}^{3 \times 512 \times 512} \to \mathbb{R}^{1024 \times d} \tag{8}$$

## C.3 Text Classification Experimental Details

The parameters used to extract gradients from text encoders for $\mathcal{L}_{\mathrm{KL}}$ and $\mathcal{L}_{\mathrm{SimCLR}}$ are shown in Table 24. The gradient source layer is always the attention output projection of the last transformer encoder block. We use the same parameters across backbones. No data augmentation is used for the $\mathcal{L}_{\mathrm{KL}}$, while for $\mathcal{L}_{\mathrm{SimCLR}}$ the views are obtained by randomly deleting words independently, where each word has a 10% probability of being deleted.

Table 23: **In-context scene understanding setup**. Parameters (left) and data augmentation (right) used for the in-context scene understanding task for both full dataset and few shot setups. For the computation of $\mathcal{L}_{\text{SimCLR}}$ we use 1025 views as we include the [CLS] token, which is discarded afterwards. The retrieved negatives indicate the number of neighbors retrieved from the support index, while the loss negatives the number of neighbors used for the loss computation. The color data augmentations are applied independently, in the order shown here.

| Parameter | Full Dataset | Few-Shots |
|---|---|---|
| Memory Bank Size | See Table 3 | $2048 \times 10^4$ |
| Nearest Neighbors $k$ | 30 | 90 |
| Temperature | 0.02 | 0.1 |
| Augmentation Epochs | 1 | 8 |
| ScaNN Index | | |
| Num Leaves | 512 | 512 |
| Num Leaves to Search | 32 | 256 |
| Reordering Num Neighbors | 120 | 1800 |
| Dimensions per Block | 4 | 4 |
| Anisotropic Quantization | 0.2 | 0.2 |
| $\mathcal{L}_{\text{SimCLR}}$ | | |
| Support Index Size | See Table 3 | $2048 \times 10^4$ |
| Projection Dim | 96 | 96 |
| Positive Views | 1025 | 1025 |
| Negatives Batch Size | 1025 | 1025 |
| SimCLR Temperature | 0.07 | 0.07 |
| Retrieved Negatives per Patch | 2 | 2 |
| Loss Negatives per Patch | 1 | 1 |

| Parameter | Value |
|---|---|
| Random crop $p$ | 1.0 |
| Scale factor | $[0.5, 2.0]$ |
| Brightness jitter $p$ | 0.5 |
| Contrast jitter $p$ | 0.5 |
| Saturation jitter $p$ | 0.5 |
| Hue jitter $p$ | 0.5 |
| Max brightness $\Delta$ | 0.1 |
| Max contrast $\Delta$ | 0.1 |
| Max saturation $\Delta$ | 0.1 |
| Max hue $\Delta$ | 0.1 |

Table 24: **Text classification experimental details.** Parameters used to extract text encoder gradients for the $\mathcal{L}_{\text{KL}}$ (left) and $\mathcal{L}_{\text{SimCLR}}$ (right) objectives.

| Parameter | Value |
|---|---|
| Temperature | 15 |
| Projection Dim | 768 |

| Parameter | Value |
|---|---|
| Positive Views | 12 |
| Negative Views | 2 |
| Projection Dim | 256 |
| Negatives Batch Size | $256 \times 2$ |
| Temperature | 0.07 |
| Word Deletion $p$ | 0.1 |

# D   Data and Models

We investigate the performance of our gradient-enhanced features on 11 image classification datasets, namely CIFAR 10 and CIFAR 100 (Krizhevsky et al., 2009), Oxford Flowers 102 (Nilsback & Zisserman, 2008), Food101 (Bossard et al., 2014), ImageNet-1K (Russakovsky et al., 2015), FGVC Aircraft (Maji et al., 2013), CUB 200-2011 (Wah et al., 2011), Oxford-IIT Pets (Parkhi et al., 2012), Stanford Cars (Krause et al., 2013), DTD Textures (Cimpoi et al., 2014) and EuroSAT (Helber et al., 2019), 5 text classification datasets: TREC (Li & Roth, 2002) in its coarse version, banking-77 (Casanueva et al., 2020), Stanford Sentiment Treebank (SST) (Socher et al., 2013) in its fine-grained version, AG news (Zhang et al., 2015; Gulli, 2005) and tweet eval (emoji) (Barbieri et al., 2018, 2020) and 2 audio classification datasets: ESC 50 (Piczak, 2015), an environmental sound classification dataset, and SpeechCommands V2 (Warden, 2018), a keyword spotting task, where the goal is to classify utterances into a predefined set of words. The datasets, their license and citations are also listed in Table 25.

We follow the evaluation protocol for each individual dataset and report the top-1 accuracy for CIFAR 10 and 100, Food101, ImageNet-1K, Stanford Cars, EuroSAT, DTD Textures, CUB 200-2011, TREC, banking-77, SST, AG news, tweet eval (emoji), ESC 50 and SpeechCommands V2, and the mean-per-class accuracy for Flowers102, FGVC Aircraft and Oxford-IIT Pets.

We use the default splits defined by torchvision or the dataset authors where possible. As EuroSAT does not explicitly define a test split, we use an 80/20 stratified split as indicated by the dataset paper. We always report metrics on the test splits, with the exception of ImageNet, for which we use the validation split.

Table 25: **Datasets.** Summary table of all datasets used in this paper, their license and citation.

| Dataset | Type | License | Citation |
|---|---|---|---|
| CIFAR 10 | Image | Unknown | Krizhevsky et al. (2009) |
| CIFAR 100 | Image | Unknown | Krizhevsky et al. (2009) |
| Stanford Cars | Image | Custom (Non Commercial) | Krause et al. (2013) |
| DTD Textures | Image | Custom (Research Only) | Cimpoi et al. (2014) |
| EuroSAT | Image | MIT | Helber et al. (2019) |
| CUB 200 (2011) | Image | Custom (Research Only, Non Commercial) | Wah et al. (2011) |
| Oxford-IIT Pets | Image | CC BY-SA 4.0 | Parkhi et al. (2012) |
| Food101 | Image | Unknown | Bossard et al. (2014) |
| FGVC Aircraft | Image | Custom (Research Only, Non Commercial) | Maji et al. (2013) |
| Flowers102 | Image | Unknown | Nilsback & Zisserman (2008) |
| ImageNet 1K | Image | Custom (Research Only, Non Commercial) | Russakovsky et al. (2015) |
| ImageNet 100 | Image | Custom (Research Only, Non Commercial) | Russakovsky et al. (2015) |
| TREC | Text | Unknown | Li & Roth (2002) |
| Banking-77 | Text | CC BY 4.0 | Casanueva et al. (2020) |
| SST | Text | Unknown | Socher et al. (2013) |
| AG News | Text | Custom (Non Commercial) | Zhang et al. (2015); Gulli (2005) |
| Tweet Eval | Text | Unknown | Barbieri et al. (2018, 2020) |
| ESC 50 | Audio | CC BY-NC 3.0 | Piczak (2015) |
| SpeechCommands V2 | Audio | CC BY 4.0 | Warden (2018) |

We evaluate FUNGI features across several architectures, pretraining strategies and model sizes. These are listed in Table 26, alongside their license, data type and citation.

# E    Compute Resources

The gradient features were extracted using a machine with a single NVIDIA A100 GPU with 40GB of VRAM. Considering the inference times listed in Table 27, replicating the k-nearest neighbor image classification results would require approximately 27 GPU hours per backbone using `float16`, which we use throughout all our experiments. As we evaluate our method across 17 vision backbones, reproducing these results would require 459 GPU hours. As for the text and audio classification experiments, they require around 3 GPU hours per backbone, for a total of 9 hours. The extracted gradient features were reused for the linear probing and clustering experiments, the former requiring 168 hours on a machine with a single AMD EPYC 7H12 CPU and the latter requiring 18 hours on a machine with a single NVIDIA A100 GPU with 40GB of VRAM.

Table 26: **Models used in the paper.** Summary table of all architectures/pretraining strategies evaluated in the paper, along with their license, citation, and implementation, if applicable.

| Model | Type | License | Citation |
|---|---|---|---|
| Masked Autoencoder | Image | CC BY-NC 4.0 | He et al. (2022); Wightman (2019) |
| AugReg | Image | Apache 2.0 | Steiner et al. (2022); Wightman (2019) |
| DeIT | Image | Apache 2.0 | Touvron et al. (2021) |
| DINO | Image | Apache 2.0 | Caron et al. (2021) |
| DINOv2 | Image | Apache 2.0 | Oquab et al. (2023) |
| CLIP | Image | MIT | Radford et al. (2021); Wightman (2019) |
| EVA-CLIP | Image | MIT | Sun et al. (2023); Wightman (2019) |
| MoCov3 | Image | CC BY-NC 4.0 | Chen et al. (2021) |
| BERT | Text | Apache 2.0 | Devlin et al. (2019); Wolf et al. (2020) |
| T5 | Text | Apache 2.0 | Raffel et al. (2020); Wolf et al. (2020) |
| SSAST | Audio | BSD 3-Clause | Gong et al. (2022, 2021) |

Table 27: **FUNGI introduces a speed overhead.** Embeddings and gradients extraction speed measured in images/second on an NVIDIA A100 GPU for a DeIT ViT-B/16 backbone. The gradients speed include the random projection step. The performance column reports the accuracy averaged across 11 datasets for the combination of a single gradient with the model embeddings. † indicates k-nearest neighbor inference on CPU.

| Features Source | Images/s | Inference Speed (samples/s)† | Performance |
|---|---|---|---|
| Embeddings | 479 | 2700 | 67.3 |
| $\nabla_{\text{KL}}$ | 344 | 2700 | 68.2 ↑0.9 |
| $\nabla_{\text{DINO}}$ | 32 | 2700 | 70.1 ↑2.8 |
| $\nabla_{\text{SimCLR}}$ | 12 | 2700 | 70.9 ↑3.6 |

Finally, additional experiments such as image retrieval, in-context learning, and ablation studies required approximately 84 hours, while the preliminary experiments for this paper required a negligible amount of compute.

