# OpenReview forum: "No Train, all Gain: Self-Supervised Gradients Improve Deep Frozen Representations"
_NeurIPS.cc/2024/Conference — NeurIPS 2024 poster_

### Official Review · Reviewer_98WT · 2024-07-10

**Soundness:** 3
**Presentation:** 4
**Contribution:** 3
**Rating:** 8
**Confidence:** 4

**Summary:**

This paper presents an unsupervised method to enhance the representations of pretrained models. The authors observe that gradient features from various self-supervised objectives are helpful to enhance k-nearest neighbor (KNN) retrieval. The proposed method, FUNGI, combines the embeddings of a pretrained model with gradients from different self-supervised objectives, followed with dimensionality reduction from PCA. Experiments on Image classification, semantic segmentation and text classification show the effectiveness of proposed algorithm.

**Strengths:**

This paper is well written and eazy to follow, with nice charaterization of background and related works. The algorithm is well motivated. The plots and tables are also well-structured with clear information. The idea to combine the gradient features from multiple objective is simple and effective. The proposed algorithm is evaluated on different pretrained models and different tasks.

**Weaknesses:**

1. My biggest concern is that the evaluation of proposed method is limited to nearest neighbor retrieval (though I agree that this task is important). As a method of feature engineering, it'd be good to test the performance of (few-shot) linear prob or K-Means clustering.

2. Lack of analysis on random projection. For example, [6] uses random Gaussian matrix to project the gradient features to low-dim and shows that it perseves inner-product with high probability. In FUNGI, the projection matrix is sampled in {-1, 1} by Bernoulli random variables. What properties would it have?

3. Some related works are not covered. There've been a number of works using gradient as features for downstream tasks, for example [1-3]. It'd be good to metion them and discuss the intuition why gradient could be used as features for KNN retrieval. The same also holds for the analysis on self-supervised learning objectives, for example [6].

[1] Gradients as Features for Deep Representation Learning, ICLR 2020.

[2] More than a toy: Random matrix models predict how real-world neural representations generalize. ICML 2022.

[3] Head2Toe: Utilizing Intermediate Representations for Better Transfer Learning. ICML 2022.

[4] A Kernel-Based View of Language Model Fine-Tuning. ICML 2023.

[5] Trak: Attributing model behavior at scale. ICML 2023.

[6] Learning by Reconstruction Produces Uninformative Features For Perception. 2024

**Questions:**

1. What's the dimension of the gradient feature? What's the time cost of random projection and PCA?

2. What's the advantage of random projection with matrix sampled from {-1, 1}, compared to random Gaussian matrix?

3. For text classification, nearest neightbors are used to select examples for in-context learning [1]. Would FUNGI be helpful in this process?


[1] What makes good in-context examples for gpt-3? ACL 2022.

**Limitations:**

The author have discussed some of the limtations in the paper. For others, please refer to "Weakness"

---

> ### Author Rebuttal · Authors · 2024-08-06
>
> ### Other evaluations (linear probing, k-means, in-context learning)
>
> We conducted further evaluations of FUNGI features for logistic regression, clustering and text classification via in-context learning. The results are discussed in the global response.
>
> ### What's the dimension of the gradient feature? What's the time cost of random projection and PCA?
>
> Assuming a ViT-B/16 backbone, the weight gradients from the last attention output projection have shape $(768, 769)$, including the bias. We project them to the same dimensionality as the embeddings, in this case $768$. The final FUNGI features (before PCA) will then have shape $768 \times (N + 1)$, where $N$ is the number of gradient features used.
>
> Performing the random projection on the GPU for a batch of $8$ samples results in an overhead of $0.017$ seconds, or $0.0002$ seconds per sample.
>
> As for the PCA, fitting it over the entire training set of EuroSAT ($18,000$ samples) requires $5.18$ seconds, while transforming the training set requires $0.17$ seconds, or $0.00001$ s/sample. For larger datasets an incremental version of PCA can be used.
>
> The reported runtimes were averaged over $1000$ runs ($100$ for the PCA fitting).
>
> ### What properties does a binary random projection have? What is the advantage?
>
> TRAK [98WT-5] uses a random projection sampled from $N(0, 1)$ to reduce the dimensionality of gradients, which, according to the Johnson-Lindenstrauss lemma, preserves inner products with high probability, given a large enough projection subspace $k$. The main result of [C] is demonstrating that this lemma holds also for matrices whose entries are sampled from either $\\{-1, 1\\}$ or $\\{-1, 0, 1\\}$.
>
> Empirically we show that this holds true by evaluating the k-NN classification accuracy on Flowers102 of raw and projected KL gradients from a DeIT ViT-S/16 backbone.
>
> | Features            | Dimensionality | Accuracy |
> | ------------------- | -------------- | -------- |
> | Raw Gradients       | 147456         | 53.53    |
> | Projected Gradients | 768            | 53.17    |
>
> _**Table 28. Binary random projections preserve euclidean distances:** k-NN classification using raw and projected gradients from a DeIT ViT-S/16 backbone. Projected gradients perform marginally worse, showing that relative euclidean distances are mostly preserved._
>
> One advantage of a random matrix with $\\{-1, 1\\}$ entries is that it can be efficiently stored as a boolean matrix. For example, the projection matrices used for ViT-B models have a shape of $(768 \times 769, 768)$, equivalent to ${\sim}1.8$ GB when encoded as `float32` and to $0.45$ GB when encoded as booleans.
>
> We also evaluate the difference in performance given by random projection matrices using different initializations, shown in Table 29, which displays the mean per-class accuracy on Flowers102 of FUNGI features extracted from a DeIT ViT 16/B backbone with respect to the projection initialization. The results show that a binary matrix has a slight advantage over Gaussian and sparse matrices, and has a lower standard deviation as more sub-features are used, but the performance gap is negligible.
>
> |                   | Binary $\\{-1, 1\\}$   | Gaussian            | Sparse           |
> | ----------------- | -------------------- | ------------------- | ---------------- |
> | Embeddings        | 56.9                | 56.9               | 56.9            |
> | $\quad$ + KL       | 59.7 $\pm$ 0.4     | **60.2 $\pm$ 0.2** | 60.0 $\pm$ 0.1 |
> | $\quad$ + KL + DINO    | **63.5 $\pm$ 0.1** | 63.2 $\pm$ 0.5    | 63.1 $\pm$ 0.2  |
> | $\quad$ + KL + DINO + SimCLR | **69.2 $\pm$ 0.6** | 69.0 $\pm$ 0.7    | 68.9 $\pm$ 0.9 |
>
> _**Table 29. The random projection initialization has little impact on performance:** comparison of the downstream accuracy of FUNGI features built with gradients projected using random matrices with different initializations. We report the mean and one standard deviation measured across three runs using the Flowers102 dataset and a DeIT ViT-B/16 backbone._
>
> PS: We assumed that for this question you were referencing [98WT-5] (TRAK) rather than [98WT-6] (Learning by Reconstruction..), as the former has the exact sentence mentioned in the review. Please correct us in case.
>
> ### Missing related works
>
> We highly appreciate the suggestion of these works, we discussed them and used some of their results to build an intuition in the global response on why/how gradient features work. We will be include them in the updated version of the paper.
>
> Building on that discussion, our work is also related to [98WT-2], which used the empirical neural tangent kernel (eNTK), i.e. the per-sample Jacobian, to benchmark an estimator of generalized risk, and also shows that kernel regression on the eNTK can achieve a performance close to fine-tuning in vision. Their work is extended by [98WT-4], which shows that eNTK can be used for prompt-based fine-tuning of masked language models, if the task exhibits kernel behavior. In comparison to their work, our method does not require the downstream task to show kernel behavior, and we expect their method to be more computationally intensive, as they compute the full model Jacobian for each sample. [98WT-3] takes inspiration from NTK and propose using gradients from task-specific losses as features for representation learning. Given a pre-trained network, they first train a linear classifier in a supervised manner and subsequently learn a linear model using both activation and gradient-based features. FUNGI differs from this work by not requiring any human labeled samples and not running any training as it simply uses the gradients as features for k-NN classification.
>
> **References**
>
> [C] Database-friendly random projections: Johnson-Lindenstrauss with binary coins. Journal of computer and System Sciences (2003).

---

> ### Comment · Reviewer_98WT · 2024-08-09
> **Acknowledgement**
>
> Thank you for the detailed reply.
>
> **About evaluation on other tasks**: Evaluation on linear probe, clustering and in-context learning are good. It's also great to see that these experiences are carefully conducted in a correct way, e.g., to choose L2-regularization strength for linear probe. From my own experience, the performance of linear probe would be tricky without a proper regularization strength.
>
> **About related work and intuition of FUNGI**: the explanation that FUNGI can be regarded as update-free fine-tuning on other objectives is interesting.
>
> **About comparison to random Gaussian matrix**: Yes I'm mentioning TRAC [5] for the projection with random Gaussian matrix. I agree that random binary matrix is more memory-efficient.
>
> **About choice of gradient**: it seems that you could choose the gradient from [attn_qkv, attn_proj, mlp.fc1, mlp.fc2] as features. From Figure 8 in the paper, gradient in the early layer seems not helpful. In practice, can you explain which part of gradient to choose in order to get best performance? Will the combination of gradients from different layers/modules be helpful (many people just use full gradient of the model, which is very expensive and might not perform very well)?
>
> Overall, I'm glad to see the response from the authors. I'd raise my score to 8 and hope athors would include these changes in the modified version (especially the intuition of using gradient feature) and release their code in the future.

---

> > ### Author Response · Authors · 2024-08-13
> >
> > Thank you for your response and for updating the rating.
> >
> > **About the choice of gradients:** correct. For our experiments we use the full gradient with respect to the weights and bias of a single layer, as this is simpler (implementation-wise) and we expect this to be more stable overall, compared to picking a subset of the gradient matrix. While picking only a subset of the gradient is an interesting direction, we did not investigate it thoroughly. Nonetheless, we can make the following observations:
> >
> > - As shown in Table 30, removing the bias gradient may lead to even better performance.
> > - As shown in Table 31, for the `qkv` layer we found that using the `v` gradients alone results in better performance compared to the full `qkv` matrix and its other components.
> >
> > Combining gradients from multiple layers is an interesting idea, thank you. We ran a small experiment and from the results displayed in Table 32, it may lead to better performances, but this is not always the case.
> >
> > Both ideas require further investigation and evaluation to make any concrete claims.
> >
> > |                           | Textures | Flowers102 |
> > | ------------------------- | -------- | ---------- |
> > | Weight Gradients w/o Bias | 59.4     | 61.0       |
> > | Weight Gradients w/ Bias  | 59.0     | 60.5       |
> >
> > _**Table 30. Removing the bias gradient improves performance:** comparison of the downstream accuracy of FUNGI features built with weight only gradients or weight and bias gradients on two datasets using an AugReg IN1K ViT-B/16 backbone._
> >
> > |     | Textures | Flowers102 |
> > | --- | -------- | ---------- |
> > | Q   | 56.7     | 52.0       |
> > | K   | 55.2     | 52.0       |
> > | V   | 59.0     | 61.2       |
> > | QKV | 58.0     | 58.3       |
> >
> > _**Table 31. The `v` of the `qkv` gradients has the best performance, and can even improve over `attn.proj` (Table 30):** comparison of the downstream accuracy of FUNGI features built with the full `qkv` gradients or one of its sub-components. The backbone was an AugReg IN1K ViT-B/16._
> >
> > |                          | Textures | Flowers102 |
> > | ------------------------ | -------- | ---------- |
> > | attn.proj                | 59.0     | 60.5       |
> > | attn.proj + attn.qkv (v) | 60.9     | 59.2       |
> >
> > _**Table 32. Combining gradients may improve performance:** comparison of the downstream accuracy of `attn.proj` FUNGI features augmented (or not) with gradient features from the `v` of the `qkv` gradients. The backbone was an AugReg IN1K ViT-B/16._
> >
> > Finally, we do plan to include the discussions that surfaced in the review in the paper, as they will definitely improve the final manuscript. Indeed, we will release all code as open source, including a plug-and-play package to extract FUNGI features from any transformer backbone, e.g. `features = fungi(model, layer='7.attn.v')`.
> >
> > We're happy to address any other feedback or provide clarifications as needed.

---

### Official Review · Reviewer_EjYW · 2024-07-14

**Soundness:** 3
**Presentation:** 4
**Contribution:** 3
**Rating:** 7
**Confidence:** 4

**Summary:**

The paper proposes a simple method named FUNGI - Features from Unsupervised Gradients, to improve the representations of Vision Transformers (ViTs). Specifically, FUNGI uses gradients from self-supervised objectives to augment the embeddings from pre-trained models. FUNGI involves three straightforward steps - (i) compute gradients at a specific layer for selected self-supervised objectives (ii) project the gradients to a lower dimension (iii) concatenate these projected features with the model’s embeddings for kNN-based classification. Extensive experiments across various datasets, pre-trained models, and tasks demonstrate the effectiveness of the proposed approach.

**Strengths:**

- **Presentation**: The paper is presented well and well-structured overall. The use of gradients for feature enhancement in ViTs is motivated well through simple analytical experiments. All the experimental setups have been clearly explained and the results are presented well, covering a wide range of pre-trained models and datasets.

- **Simplicity**: The proposed method is simple and easy to understand and implement. Moreover, the authors have provided the PyTorch pseudocode in the supplementary, which makes it easier to replicate the results of the proposed method.

- **Results**: The results from Fig. 5, Tables 1, 2, 3, and 5 demonstrate the effectiveness of FUNGI across various datasets, pre-trained models, and tasks (visual and text-based). Given that the improvements come with no training, they are noteworthy.

**Weaknesses:**

- **Missing backbones**: While the authors have presented results on a comprehensive list of backbones, a few recent backbones are missing, which are listed below:
    - Touvron, Hugo, Matthieu Cord, and Hervé Jégou. "Deit iii: Revenge of the vit." ECCV 2022.
    - Vasu, Pavan Kumar Anasosalu, et al. "Mobileclip: Fast image-text models through multi-modal reinforced training." CVPR 2024.

- **Scalability**: There is no discussion on how FUNGI scales with larger ViT models. All the analyses are performed with ViT-B and ViT-S. The authors should also demonstrate the scalability of FUNGI with larger ViT models across the various pre-training schemes that they have presented results on for ViT-B and ViT-S.

**Questions:**

1. Does FUNGI also work for CNNs and CNN-ViT hybrid architectures? For example, MobileCLIP mentioned above consists of convolutional layers and attention-based layers. Would FUNGI work for CNNs such as CLIP ResNet-50 and CNN-ViT hybrids such as MobileCLIP?

2. Since FUNGI is a generic gradient-based method, does it extend beyond classification tasks? Would FUNGI work for generative models such as LLMs, VLMs such as Llava, and BLIP? The application of FUNGI to any such model would greatly improve the impact of the work.

**Limitations:**

The authors have adequately addressed the limitations of the work.

---

> ### Author Rebuttal · Authors · 2024-08-06
>
> ### Missing backbones
>
> We extend the evaluation of FUNGI to DeiT-III, MobileCLIP and CLIP and MoCov2 ResNet-50 backbones. For MobileCLIP, we extract gradient features from the attention output projection of the last token mixer, and for ResNet-50 models we use the last convolutional layer, i.e. `layer4.2.conv3`. For ResNet-50 models we only evaluate KL and DINO gradients, as SimCLR gradients did not seem particularly promising.
>
> Although we find that our method works consistently with ViTs, the results on CNNs or ViT-CNN hybrids are less consistent. While our method improves on MoCov2 ResNet-50, it does not improve on the CLIP ResNet-50 and MobileCLIP-S1. We would like to stress that these are preliminary results, and different data augmentation strategies or a more throughout evaluation of the possible gradient sources (i.e. layers) may yield better results for CNNs or ViT-CNNs. The results, averaged over 11 datasets, are reported in Table 26 and Table 27 for full dataset and few shot setups respectively.
>
> |                      | DeiT III ViT-B/16 | CLIP ResNet50 | MoCov2 ResNet50 | Mobile CLIP S1 |
> | -------------------- | ----------------- | ------------- | --------------- | -------------- |
> | Embeddings           | 64.2              | **65.7**      | 52.7            | **79.8**       |
> | + KL                 | 64.8  ↑0.6        | 63.9   ↓1.8   | 52.6 ↓0.1       | 78.2 ↓1.7      |
> | + KL + DINO          | 67.3  ↑3.1        | 63.7  ↓2.0    | **53.6 ↑0.9**   | 78.1 ↓1.8      |
> | + KL + DINO + SimCLR | **68.2 ↑4.0**     | --            | --              | --             |
>
> _**Table 26. FUNGI works primarily for ViTs:** accuracy in full dataset k-nearest neighbor evaluation averaged over 11 datasets for FUNGI and embeddings._
>
> |                      | DeiT III ViT-B/16 | CLIP ResNet50 | MoCov2 ResNet50 | Mobile CLIP S1 |
> | -------------------- | ----------------- | ------------- | --------------- | -------------- |
> | Embeddings           | 34.9              | **34.7**      | 26.4            | **47.9**       |
> | + KL                 | 36.2  ↑1.3        | 33.4 ↓1.3     | 26.8 ↑0.4       | 46.2 ↓1.7      |
> | + KL + DINO          | 37.2  ↑2.3        | 32.5 ↓2.2     | **27.6 ↑1.2**   | 46.6 ↓1.3      |
> | + KL + DINO + SimCLR | **39.6 ↑4.7**     | --            | --              | --             |
>
> _**Table 27. FUNGI works primarily for ViTs:** accuracy in few shot k-nearest neighbor evaluation averaged over 11 datasets for FUNGI and embeddings._
>
> ### Scalability
>
> In Table 18 of the uploaded PDF we report results for AugReg (labelled as "AR"), CLIP and DINOv2 ViT-L models, and a DeiT ViT-H/14 model, for full dataset and few shot setups. The results are averaged over 11 datasets, except for DeiT ViT-H/14, whose results exclude CIFAR 10 and 100, Food-101 and ImageNet-1K due to computational reasons.
>
> The results show that FUNGI works also for larger backbones, especially in few-shot setups, where it improves by $+4.4\\%$, $+4.3\\%$ and $+2.1\\%$ for CLIP ViT-L/14, DINOv2 ViT-L/14 and DeiT ViT-H/14 respectively.
>
> ### Does FUNGI extend beyond classification tasks? Can it be used for generative models such as LLMs, VLMs such as Llava, and BLIP?
>
> We're thankful for this suggestion, and yes, in the global response we show that FUNGI can be used to improve the performance of linear probing, clustering and that it can be used to retrieve better example for in-context learning with LLMs.
>
> As for integrating FUNGI in the pipeline of generative models such as BLIP or LLaVA we believe it should be possible, as they both use a frozen vision encoder. But, taking BLIP as an example, the vision features are used by the text decoder via cross-attention, and thus we expect FUNGI features to be out-of-distribution without at least a partial re-training or fine-tuning. We did not have enough time to setup such an experiment during the rebuttal, but we agree that it would be an interesting direction to explore.

---

### Official Review · Reviewer_8XpZ · 2024-07-18

**Soundness:** 2
**Presentation:** 3
**Contribution:** 3
**Rating:** 4
**Confidence:** 4

**Summary:**

- The draft introduces a feature enhancement technique called FUNGI (Features from Unsupervised GradIents) for vision transformers. -
- The core idea is to leverage the un/self-supervised loss gradients at an arbitrary hidden layer within a vision backbone (the default option being the attention output projection of the last transformer block) to enhance the feature embeddings
- These gradients, along with the pre-trained embeddings, are shown to achieve higher accuracy in kNN classification across various datasets and transformer backbones.
- Comprehensive experimentation evaluated this technique in three activities (image classification, in-context scene understanding, and text classification). Consistent gains in accuracy are reported across model backbones, tasks/datasets, learning scenarios, etc.

**Strengths:**

1. The core contribution is to show that gradients w.r.t. self-supervised objectives have complementary information to the pre-trained model embeddings. This information can be combined to achieve better performance. In my opinion, this can be claimed to be a novel representation.

It is well-known that pretraining on self-supervised objectives results in moderately powerful features. Contrary to this notion, the draft presents a method to enrich the trained embeddings with more information from multiple such self-supervision objectives. However, the clever thing the draft does is to avoid costly fine-tuning by concatenating the (one-step) gradients for the learned embeddings toward the self-supervision tasks.

2. The authors have introduced a "generic" technique to enhance the performance of the kNN classifier built on top of a transformer backbone.
3. The experimental evaluation of the proposed technique is sound, with several backbones, various unsupervised objectives, different tasks/datasets, ablations, initial verifications (Figures 2 & 3), etc.

**Weaknesses:**

1. Although complementary information is extracted as gradients, it is computationally demanding - it requires one (partial) backpropagation for each self-supervision objective, total backpropagation operations are linear with respect to the number of objectives, the added linear projection head (h), and finally, the dimensionality reduction to match the dimension of the embedding. One may opine that the additional cost is not worth performance gains (poor returns). Overall, the gains may not be significant to be deployed in practice.

2. The draft provides no sound reasoning regarding the performance enhancement resulting from the self-supervised gradients. Section 3 (figures 2 & 3) empirically proves that these gradients complement the embeddings but doesn't discuss how/why.

**Questions:**

1. It would be interesting to understand (if not already) the role of the pre-trained embedding. In other words, can a half-trained or untrained (random) embedding also extract discriminative information from the self-supervised objectives similar to the pre-trained one? Authors may clarify this.

2. Authors may clarify the second weakness mentioned in the above section.

**Limitations:**

The authors addressed the limitations and potential societal impact of the proposed work.

---

> ### Author Rebuttal · Authors · 2024-08-06
>
> ### Computational cost
>
> Our method does indeed introduce a computational overhead, but we believe that in a retrieval setting, where the embeddings for the database are computed once and only query samples are encoded on the fly, the performance improvement may be very well worth the added computational cost.
>
> In particular, on retrieval on the revisited Paris landmarks dataset [B], our KL loss improves the mean average precision (mAP) of a CLIP ViT-B/16 backbone by $+10.1\\%$ and $+12.4\\%$ on the medium and hard splits respectively, at the cost of a ${\sim}28\\%$ reduction in throughput, including the random projection. The results are also shown in Table 24.
>
> |          | Medium         | Hard           |
> | -------- | -------------- | -------------- |
> | Features | 64.6           | 40.4           |
> | $\quad$ + KL     | **74.7 ↑10.1** | **52.8 ↑12.4** |
>
> _**Table 24. FUNGI improves retrieval:** retrieval mAP on the Paris landmarks dataset of embeddings and FUNGI features built with KL gradients using a CLIP ViT-B/16 backbone._
>
> Beyond that, we note that the main scope of this work was to demonstrate that gradient-enhanced features can improve performance across several tasks, modalities, and backbones. We acknowledge that our method, particularly with the DINO and SimCLR losses, introduces a computational overhead due to the requirement of multiple views for each individual sample. However, we believe that it is possible to make the computation of these gradient features significantly more efficient, which we leave for future work.
>
> ### Why are gradients and embeddings complementary?
>
> In the global response we built some further intuition on why SSL-gradient features can improve the pretrained embeddings. Moreover, the second row of Figure 2 in the main paper shows that gradient alone can already provide a significant performance improvement, thus it's not strictly necessary to combine them with the embeddings.
>
> On the other hand, gradient features may be "brittle", as gradients are estimated using a single data point and depend on the local curvature of the loss landscape. Thus, we hypothesize that combining gradients and embeddings may address this issue and provide more stable yet discriminative features, which also result in better performance overall.
>
> ### What is the role of pre-trained embeddings? Can random or half-trained embeddings be used to extract discriminative information?
>
> Thank you for this interesting idea and question, that can bring new insights into the utility of gradient features. Indeed, it is possible to extract discriminative information from both a half-trained and random embeddings. For the former, as we do not have access to a half-trained backbone, we consider a Masked Autoencoder ViT-B/16, which is known to have bad performance in retrieval and requires (almost) full-fine tuning for peak performance, while for the latter we consider a randomly initialized ViT-B/16.
>
> In Table 25 we show the performance of embeddings and FUNGI features for these backbones, averaged over 11 datasets. The results show that it's indeed possible to extract discriminative information from such embeddings, in particular we improve over a random embedding by $6.25\times$ and by $1.89\times$ on MAE embeddings.
>
> |                   | MAE ViT-B/16       | Random ViT-B/16    |
> | ----------------- | -------------- | -------------- |
> | Embeddings        | 24.0           | 2.5            |
> | $\quad$ + KL       | 44.4 ↑20.4     | 12.8 ↑10.3     |
> | $\quad$ + KL + DINO    | **45.4 ↑21.4** | 14.0   ↑11.5   |
> | $\quad$ + KL + DINO + SimCLR | 38.8 ↑14.8     | **15.6 ↑13.1** |
>
> _**Table 25. FUNGI significantly improves retrieval for random or half-trained backbones:** average accuracy, over 11 datasets, in full dataset k-nearest neighbor classification for a MAE and a randomly initialized backbone._
>
>
> **References**
>
> [B] Revisiting Oxford and Paris: Large-scale image retrieval benchmarking. CVPR 2018.

---

### Official Review · Reviewer_gxYR · 2024-07-23

**Soundness:** 3
**Presentation:** 2
**Contribution:** 3
**Rating:** 6
**Confidence:** 3

**Summary:**

The paper introduces an approach to augment the feature representations from ViTs, by utilizing the gradients from self-supervised losses. The gradients from the attention output projection of the last transformer block is extracted and projected into the output embedding space using random-projections and PCA to compute FUNGI. Exhaustive experimental evaluation across vision and NLP datasets brings out the efficacy of the approach.

**Strengths:**

+ Novel methodology
+ Simple approach
+ Exhaustive experimental analysis

**Weaknesses:**

- The method to generate features from unsupervised gradients seems very empirical. Any theoretical backing for the good results would further improve the paper.

**Questions:**

- Intuitively, the proposed approach "encodes first-order learning signals that are specific to the downstream data" (line 46). If we use task specific loss functions, instead of self-supervised loss, would it further help with the adaptation of the latent representation towards the downstream task?
- How much is the method dependent on the PCA that is performed to reduce the dimensionality (line 141)? Would there be any other alternative to try?
- FUNGI has been found effective is augmenting features used for k-NN based classification and scene understanding task. Can FUNGI features be consumed by downstream discriminative classifiers that are trained with softmax?
- Would be good to clarify whether the reported numbers are the mean of multiple runs.

---

> ### Author Rebuttal · Authors · 2024-08-06
>
> ### Can task specific loss functions help with adaptation?
>
> In this work, we only use self-supervised losses that do not need human labels. If we understand correctly the reviewer's question, task-specific losses such as cross-entropy or a ranking loss are supervised and would require us to have access to the labels. Moreover, if that were the case, we could easily achieve perfect accuracy, as the label information would "leak" into the gradient features, as they are calculated on a per-sample basis.
>
> On the other hand, different losses exhibit different properties, e.g. DINO, being a clustering loss, is particularly helpful in improving the k-means clustering performance, as shown in Table 21 of the uploaded PDF. In particular, using FUNGI features from a CLIP ViT-L/14 backbone, we can improve the cluster/class overlap by up to $+15\\%$.
>
> ### How much is the method dependent on the PCA? Can other alternatives be used?
>
> The primary role of PCA in our method is to produce features that have the same storage and compute requirements (for retrieval) as the model embeddings. In Table 11 of the appendix (reported here as Table 22 for convenience) we compare the performance of PCA-transformed features to raw features, and find that, on average across 11 datasets, PCA provides a consistent minor performance boost (up to $+0.3\\%$), but it's not essential for good performance of FUNGI features.
>
> |                              | No PCA   | PCA           |
> | ---------------------------- | -------- | ------------- |
> | Embeddings                   | 65.1     | 65.3 ↑0.2     |
> | $\quad$ + KL                 | 66.0     | 66.3 ↑0.3     |
> | $\quad$ + KL + DINO          | 67.8     | 68.1 ↑0.3     |
> | $\quad$ + KL + DINO + SimCLR | **69.8** | **70.1 ↑0.3** |
>
> _**Table 22. PCA slightly improves performance, but it's not essential:** performance of FUNGI features and embeddings from a DeIT ViT-B/16 backbone transformed (or not) using PCA, averaged over 11 vision datasets._
>
> As for the strategy used for dimensionality reduction, in Table 23 we evaluate the performance of using no reduction, PCA and random projections with different initializations, and find PCA to be the best across the board.
>
> |                      | No Reduction    | PCA                 | Rand Proj (Binary) | Rand Proj (Gaussian) | Rand Proj (Sparse) |
> | :------------------- | :-------------- | :------------------ | :----------------- | :------------------- | :----------------- |
> | Embeddings           | 57.2 $\pm$ 0.0 | **61.6 $\pm$ 0.0** | 55.5 $\pm$ 0.4    | 55.8 $\pm$ 1.0     | 56.0 $\pm$ 0.5   |
> | + KL                 | 59.4 $\pm$ 0.0 | **64.0 $\pm$ 0.0**  | 59.0 $\pm$ 0.8   | 58.2 $\pm$ 0.8      | 58.7 $\pm$ 0.3    |
> | + KL + DINO          | 64.3 $\pm$ 0.0 | **68.3 $\pm$ 0.0** | 62.9 $\pm$ 0.4   | 62.2 $\pm$ 0.3      | 61.9 $\pm$ 0.5   |
> | + KL + DINO + SimCLR | 69.2 $\pm$ 0.0 | **70.9 $\pm$ 0.0** | 67.7 $\pm$ 0.5    | 67.0 $\pm$ 0.4     | 66.9 $\pm$ 0.7   |
>
> _**Table 23. PCA is the best dimensionality reduction method:** mean per-class accuracy on Flowers102 of embeddings and FUNGI features from a DeIT ViT-B/16 backbone transformed with different dimensionality reduction methods. We report the mean performance and one standard deviation across three seeds._
>
> ### Can FUNGI features be consumed by downstream discriminative classifiers that are trained with softmax?
>
> Yes, we conducted a thorough evaluation of FUNGI features in logistic regression. See the global response for details.
>
> ### Are the reported numbers the mean of multiple runs?
>
> The numbers reported in the paper refer to a single run, except for Table 16 in the appendix, where, for 8 datasets and a DeIT ViT-B/16 backbone, we report the mean accuracy over three runs and one standard deviation, which is generally low, falling below $0.3$ in most cases, and being $0.6$ at most, indicating that our method provides consistent results.

---

### Author Rebuttal · Authors · 2024-08-06

We thank the reviewers for the constructive feedback. In this general response, we address comments, shared by multiple reviewers, regarding the extension of FUNGI evaluation beyond k-NN classification and providing further insight into why FUNGI improves performance. Then, reviewer-specific responses address individual comments. The uploaded PDF includes tables with additional experimental results requested by the reviewers.

We mark references to papers cited by reviewers as `[reviewer-index]` and ours using letters, e.g. `[A]`.

### Beyond K-NN classification evaluation (gxYR, EjYW, 98WT)

**Linear probing.** We evaluate FUNGI features with logistic regression using the cyanure library [A], for all vision (backbone, dataset) pairs. Each classifier is trained for up to 300 epochs (30 in case of ImageNet) using $L_2$ regularization. For each feature combination we pick the best regularization hyper-parameter between 5 linearly spaced values in the interval $[5 \times 10^{-6}, 5 \times 10^{-4}]$ on the validation set. For datasets without a validation set, we generate one from the training set using an $80/20$ stratified split. The results are shown in the Tables 19 and 20 of the uploaded PDF. FUNGI improves linear classification in nearly all cases, especially with supervised backbones. The only exception is with DINO and DINOv2 ViT-B backbones in the few-shot setting, where FUNGI decreases performance.

**Clustering.** We evaluate FUNGI features from DeiT ViT-B/16, CLIP ViT-L/14 and DINOv2 ViT-L/14 backbones in k-means clustering. For this task, we only use the DINO and KL losses, as the SimCLR gradients did not yield good performance improvements. The clusterings are evaluated by matching clusters and classes via the Hungarian algorithm, and measuring the average (cluster, class) overlap. The results are reported in Table 21 of the uploaded PDF, and show that FUNGI can significantly improve performance, by up to $+10.8\\%$ in the case of DINOv2 ViT-L/14 and by up to $+15.8\\%$ on CLIP ViT-L/14.

**In-Context Learning.** We thank reviewers `EjYW` and `98WT` for suggesting the application of FUNGI to generative models and LLMs. In particular, reviewer `98WT` suggested that FUNGI could be used to enhance the example selection for language in-context learning (ICL). Thus, we ran a small experiment for intent classification using the banking-77 dataset (where a banking-related intent must be classified within $77$ possible classes) using $\texttt{gpt-4o-mini}$ as the LLM backbone.

For each test sample, we retrieve the top-20 most similar training set examples, and append them to the prompt alongisde their label. We then ask the model to predict the label for the test sample. For a fair evaluation, we set the model temperature to $0$.

Using the model embeddings to retrieve the ICL examples results in an accuracy of $88.7\\%$, while using FUNGI features (built with KL and SimCLR gradients) results in a $91.2\\%$ accuracy, an improvement of $+2.5\\%$.

For reference, we used the following prompt template:

```
You have to annotate banking-related queries with an appropriate intent.
You must choose a single class in the following comma-separated list:

{list of possible labels, comma-separated}

You must reply only with the class name, nothing more. Here's some examples:

{(text, label) pairs from the training set}

The query sample is: {query text}
```

labels are given as strings, e.g. `exchange_rate`.

### Theoretical backing and intuition on gradient contribution (gxYR, 8XpZ)

While we cannot provide a sound proof that theoretically explains why our method works, we can build some intuition using some papers that reviewer `98WT` pointed out, for which we're thankful for.

Similarly to our approach, [98WT-1] utilizes gradients, obtained by calculating the Jacobian for multiple deep layers, alongside the model activations to train a linear classifier with improved performance. They frame their method as a linear approximation around the model parameters, and argue that it approximates fine-tuning. Similarly, one of the core assumption that motivates [98WT-3] is that fine-tuning can be approximated as a Taylor expansion of the form:

$$
F(x,w*) \approx F(x, w) + \sum_{i,j} \frac{\delta F(x, w)}{\delta w_{i,j}} \Delta w_{ij}.
$$

Considering this, our method can be interpreted as a form of update-free fine-tuning, that gives rise to different model abilities depending on the loss being used, e.g. SimCLR, an instance discrimination objective, excels in retrieval, resulting in performance improvements of up to $+11\\%$ in k-NN classification (DeiT ViT-B/16, Flowers102), and DINO, a clustering loss, can improve the matching between k-means clusters and the original classes by up to $+15.8\\%$ (CLIP ViT-L/14, Pets).

On the other hand, pixel reconstruction losses such as masked autoencoding (MAE), which have little alignment with semantics [98WT-6], produce uninformative gradients. For example, using a MAE loss on a MAE ViT-B/16 backbone results in a ${\sim}30\\%$ top-1 accuracy on CIFAR-10, compared to an accuracy of $51.43\\%$ using activations.


**References**

[A] Cyanure: An open-source toolbox for empirical risk minimization for python, c++, and soon more. arXiv preprint 2019.

---

> ### Author Response · Authors · 2024-08-13
>
> We’re thankful to the reviewers for their comments. We’re happy to answer any further questions or clarify our responses as needed.

---

### Decision · Program_Chairs · 2024-09-25

**Decision:**

Accept (poster)

**Comment:**

Overall, the paper is well-received with three positive reviews and one borderline reject. The reviewers commend the paper's clarity, novelty, and effectiveness. While some concerns were raised regarding missing backbones, scalability, and theoretical explanations, the paper's strengths outweigh these weaknesses. The authors' ability to demonstrate the effectiveness of their proposed method across various datasets and pre-trained models is particularly noteworthy. The paper's contribution of combining self-supervised gradients with pre-trained embeddings is novel and promising.